# The role of microglia and their CX3CR1 signaling in adult neurogenesis in the olfactory bulb

Ronen Reshef[1], Elena Kudryavitskaya[2,3], Haran Shani-Narkiss[2,3], Batya Isaacson[4], Neta Rimmerman[1], Adi Mizrahi[2,3], Raz Yirmiya[1]*

[1]Department of Psychology, The Hebrew University of Jerusalem, Jerusalem, Israel; [2]Department of Neurobiology, Institute for Life Sciences, The Hebrew University of Jerusalem, Jerusalem, Israel; [3]The Edmond and Lily Safra Center for Brain Sciences, The Hebrew University of Jerusalem, Jerusalem, Israel; [4]Department of Immunology and Cancer Research, The Lautenberg Center for General and Tumor Immunology, The Hebrew University of Jerusalem, Jerusalem, Israel

**Abstract** Microglia play important roles in perinatal neuro- and synapto-genesis. To test the role of microglia in these processes during adulthood, we examined the effects of microglia depletion, via treatment of mice with the CSF-1 receptor antagonist PLX5622, and abrogated neuronal-microglial communication in CX3C receptor-1 deficient ($Cx3cr1^{-/-}$) mice. Microglia depletion significantly lowered spine density in young (developing) but not mature adult-born-granule-cells (abGCs) in the olfactory bulb. Two-photon time-lapse imaging indicated that microglia depletion reduced spine formation and elimination. Functionally, odor-evoked responses of mitral cells, which are normally inhibited by abGCs, were increased in microglia-depleted mice. In $Cx3cr1^{-/-}$ mice, abGCs exhibited reduced spine density, dynamics and size, concomitantly with reduced contacts between $Cx3cr1$-deficient microglia and abGCs' dendritic shafts, along with increased proportion of microglia-contacted spines. Thus, during adult neurogenesis, microglia regulate the elimination (pruning), formation, and maintenance of synapses on newborn neurons, contributing to the functional integrity of the olfactory bulb circuitry.
DOI: https://doi.org/10.7554/eLife.30809.001

*For correspondence:
razyirmiya@huji.ac.il

**Competing interests:** The authors declare that no competing interests exist.

## Introduction

Over the past decade it became evident that microglia play crucial roles in the regulation of synaptic processes (*Tremblay et al., 2011*; *Wake et al., 2013*), with important implications to normal behavior and to neuro- and psycho-pathology (*Yirmiya and Goshen, 2011*; *Prinz and Priller, 2014*; *Chung et al., 2015*; *Yirmiya et al., 2015*). During the early postnatal brain development period, microglia were found to be important for synapse development. For example, in the developing retino-geniculate system, microglia in the dorsal Lateral Geniculate Nucleus (dLGN) prune pre-synaptic terminals of the incoming retinal ganglion cells axons, contributing to the segregation of inputs into eye-specific territories (*Schafer et al., 2012*). One molecular system underlying neuronal-microglial interactions involves the microglial CX3C receptor-1 (CX3CR1) and its neuronally-derived CX3C Ligand-1 (CX3CL1) (*Jung et al., 2000*; *Paolicelli et al., 2014*; *Pagani et al., 2015*) During early postnatal development, CX3CR1 signaling plays a critical role in synaptic pruning. Specifically, mice with $Cx3cr1$ deletion ($Cx3cr1^{-/-}$ mice) exhibit reduced dendritic spine pruning, abnormal synapse maturation, decreased functional connectivity and behavioral abnormalities (*Paolicelli et al., 2011*; *Zhan et al., 2014*). During adulthood, microglia are involved in pruning of synapses on mature neurons. For example, following sensory (olfactory) deprivation, dopaminergic

neuron synapses are eliminated from the olfactory bulb (OB) via a mechanism that involves altered interactions between neurons and microglial processes (*Grier et al., 2016*). Moreover, microglial depletion induces an increase in spine density in the hippocampal CA1 area (*Rice et al., 2015*).

In the adult brain, new neurons are constantly added to the hippocampus and the OB circuits (*Lledo et al., 2006*; *Gould, 2007*). In the OB, this developmental process was found to be accompanied by highly dynamic synapse formation and pruning (*Mizrahi, 2007*; *Livneh et al., 2009*; *Livneh and Mizrahi, 2012*). This remodeling process allows newborn OB cells to influence neuronal circuits and behavior. For example, adult-born Granule Cells (abGCs) inhibit and sharpen the responsiveness of mitral cells (the principal OB output cells) to odors (*Egger and Urban, 2006*), and contribute to odor discrimination (*Abraham et al., 2010*).

Microglia were found to play a global role in adult neurogenesis (*Ziv and Schwartz, 2008*; *Ekdahl, 2012*; *Gemma and Bachstetter, 2013*; *Sierra et al., 2014*), by phagocytosing apoptotic newborn cells under normal physiological conditions (*Sierra et al., 2010*), facilitating the increased neurogenesis induced by rearing in an enriched environment (*Ziv et al., 2006*; *Maggi et al., 2011*; *Reshef et al., 2014*), and decreasing neurogenesis following exposure to inflammatory challenges (*Ekdahl et al., 2003*; *Monje et al., 2003*; *Lazarini et al., 2012*). A recent study also demonstrated that interactions between microglia and adult-born neurons in the OB mediate the elimination of adult-born neurons and pruning of their synapses following deafferentation-induced sensory deprivation (*Denizet et al., 2017*). However, the role of microglia in the formation, pruning, maturation, and output of adult-born neuronal synapses under normal, physiological conditions is unknown. To test the causal role of microglia in adult neurogenesis we used two approaches for modulating microglial functioning, firstly by inducing a near complete depletion of microglia, via chronic treatment of mice with the CSF-1 receptor antagonist PLX5622 (*Elmore et al., 2014*; *Dagher et al., 2015*) and secondly by studying the effects of impaired microglial-neuronal interactions in $Cx3cr1^{-/-}$ mice.

## Results

### Microglia depletion reduces spine density in adult-born neurons

In the adult neurogenesis process, new neurons are constantly generated in the subventricular zone, migrate into the OB via the rostral migratory stream (RMS), and upon their arrival begin to integrate into the existing neural circuits. To label abGCs, we infected neuroblasts migrating to the OB by injecting adeno-associated virus-1 (AAV-1), encoding tdTomato (red fluorescent protein), into the RMS (*Figure 1a*). The virus was injected to mice just before the initiation of treatment with PLX5622 (Plexxikon Inc., U.S.A.) - a selective antagonist of colony stimulating factor receptor-1 (CSFR1). In the brain, this receptor is exclusively expressed by microglia and its blockade induces microglial depletion. The depletion is already present following 7 days of treatment and is near complete by 21 days (*Elmore et al., 2014*; *Dagher et al., 2015*). Indeed, continuous administration of PLX5622 for 25 days caused dramatic microglia depletion in the OB (*Figure 1c–d*). Specifically, microglia numbers decreased from an average (±S.E.M.) density of 271 (±11) microglia cells per mm$^2$ in controls to 60 (±2) in PLX5622 -treated animals. A similar microglial depletion was also present in other areas of the CNS (*Figure 1—figure supplement 1*). The microglial depletion was also evident in the transcriptional level, as revealed by a whole transcriptome RNA-seq analysis of OBs isolated from PLX5622-treated and their respective control mice. A total of 131 genes were differentially regulated in the PLX5622 mice, of which 127 genes were down-regulated ($p<0.01$, with a cutoff of ±1.35 fold change, *Supplementary file 1*). Of the 127 down-regulated gene transcripts, 112 have been previously identified as microglia unique or enriched (*Hickman et al., 2013*; *Butovsky et al., 2014*), and likely reflect the direct consequences of microglial depletion. We examined the effects of microglial depletion on the density and size of spines on the distal apical dendrites of abGCs within the external plexiform layer (EPL), where GCs form reciprocal synapses with mitral cells (MCs) - the bulb's main projection neurons (*Figure 1b*).

In the first experiment, PLX5622 was continuously administered via the diet from 3 to 28 days post injection (d.p.i) (*Figure 2a*). The overall neurogenesis process in the PLX-5622-treated mice was similar to that in controls (*Figure 2—figure supplement 1a–b*). The number of spiny dendrites per cell (*Figure 2—figure supplement 1c*) and the average length of these dendrites/cell were

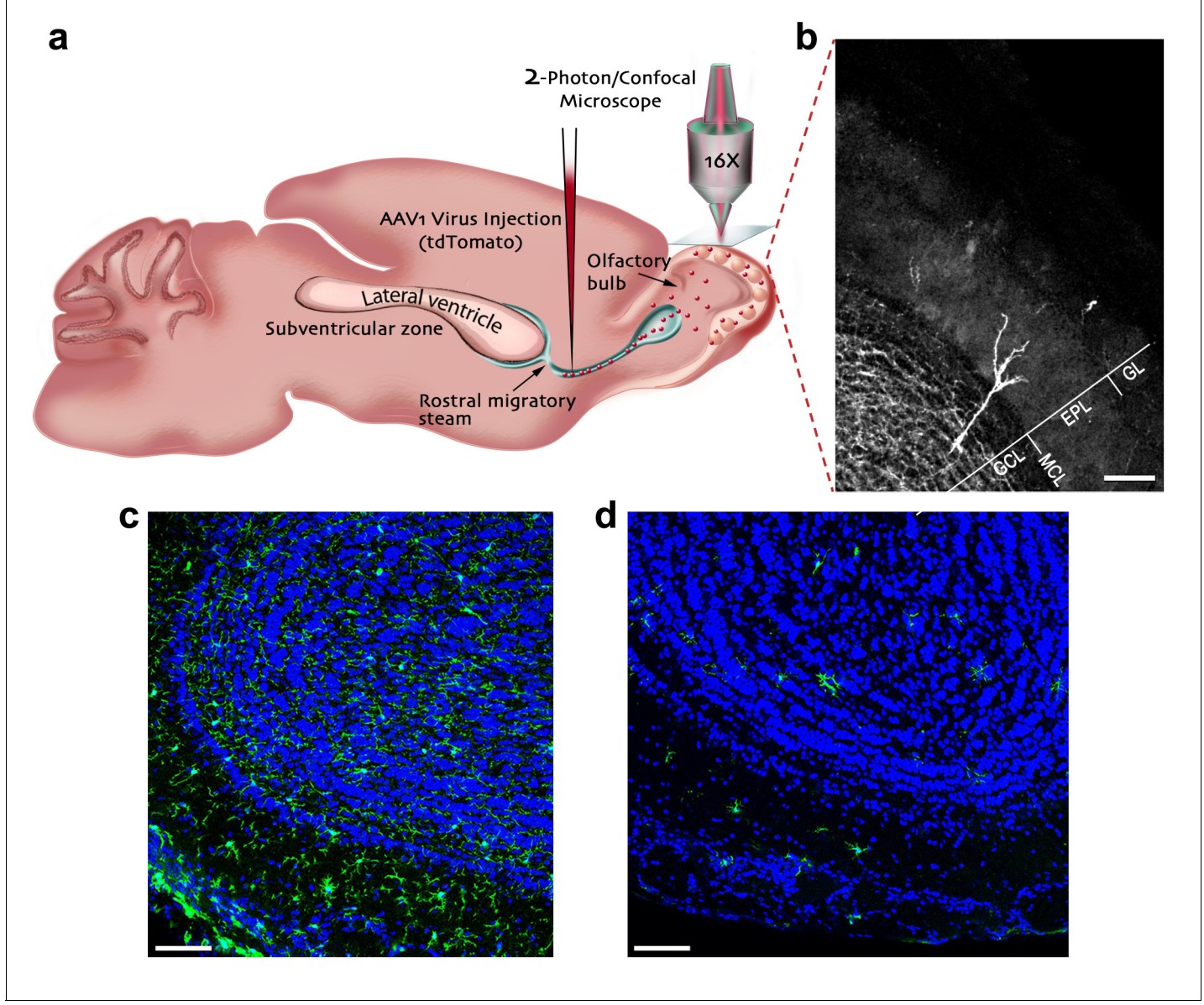

**Figure 1.** Experimental preparation for investigating the role of microglia in the development, maturation and plasticity of adult-born granule cells in the olfactory bulb. (a) A scheme of the experimental design: Mice were injected into the rostral migratory stream (RMS) with a virus that induces the expression of TdTomato fluorescent protein in adult-born neurons migrating from the subventricular zone (SVZ) to the OB. (b) A micrograph depicting an adult-born granule cell, expressing TdTomato, with its cell body in the granule cell layer (GCL) and its distal dendrites extending within the external plexiform layer (EPL; situated under the glomerular layer (GL)). Scale bar: 100 μm. (c) Fluorescent micrograph of a coronal section from the OB in a mouse fed with control diet shows the distribution of Iba-1 labeled microglia (green ells) within the OB (blue = DAPI nuclear staining). (d) Fluorescent micrograph of a coronal section from the OB in a mouse fed with PLX5622-containing diet for 28 days, demonstrating a near complete depletion of microglia in the OB. Scale bar: 100 μm.

DOI: https://doi.org/10.7554/eLife.30809.002

The following figure supplement is available for figure 1:

**Figure supplement 1.** PLX5622 treatment induces brain-wide microglia depletion.

DOI: https://doi.org/10.7554/eLife.30809.003

comparable (*Figure 2—figure supplement 1d*). The number of young new-born neurons (labeled by DCX) was also comparable between the PLX5622-treated animals and controls (*Figure 2—figure supplement 2a–d*). The number of cells co-labeled by BrdU and NeuN was also similar between the PLX5622-treated and the control mice (*Figure 2—figure supplement 3a–d*). Because BrdU was

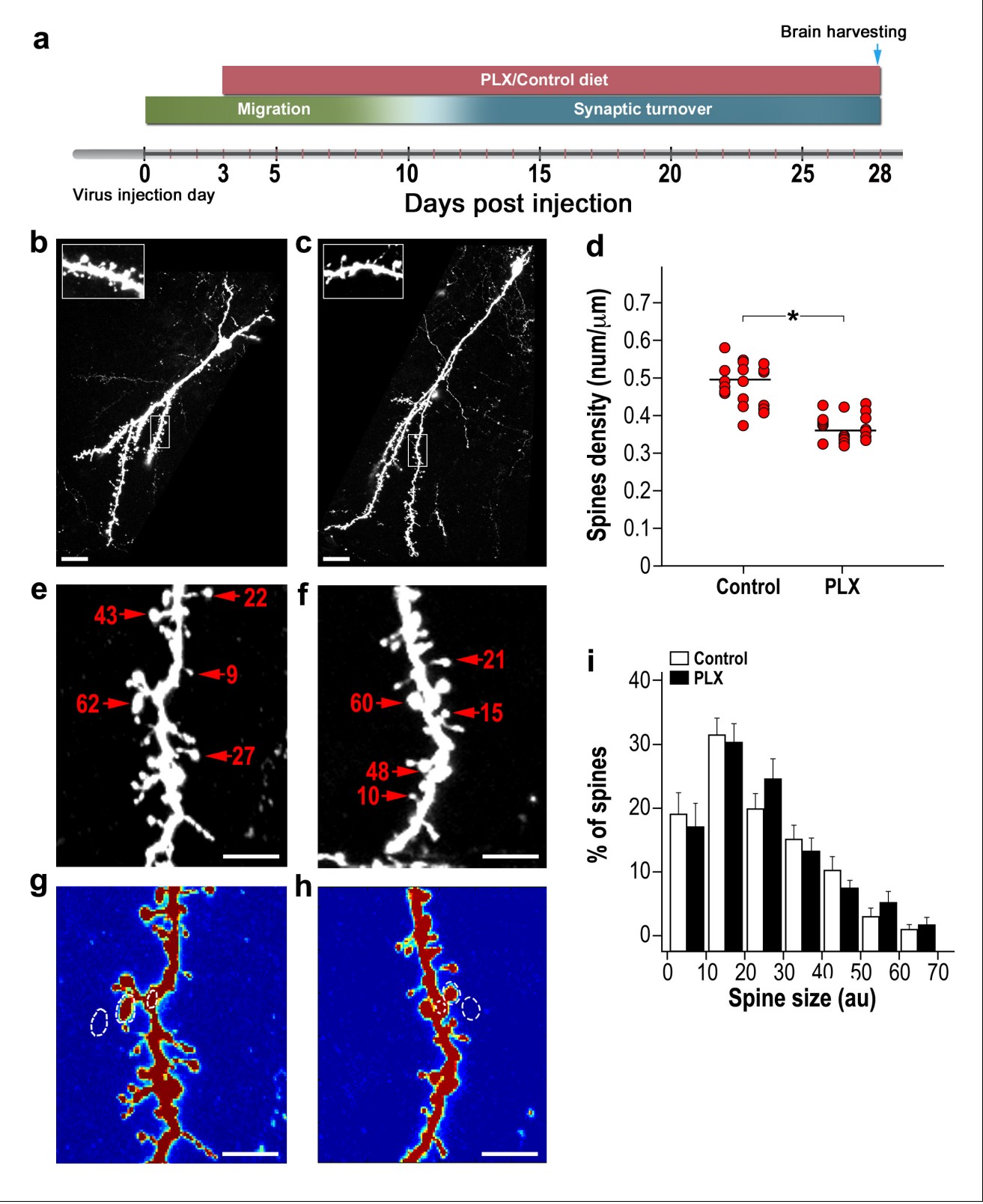

**Figure 2.** Adult-born granular cells (abGCs) in microglia-depleted mice have lower spine density, but not spine size, compared to abGCs in WT control mice. (a) Schematic time-line of the experiment. Adult-born cells were transduced by an injection of Td-Tomato-expressing adeno-associated virus 1 (AAV1) into the RMS. PLX5622/control diet feeding commenced 3 days later. Brains were harvested 28 days after the AAV1 injection. (b) High resolution projection images of abGCs and their spiny dendritic branches from a control mouse and (c) a microglia-depleted (PLX5622-treated) mouse. Scale bar:

*Figure 2 continued on next page*

*Figure 2 continued*

20 μm. Insets: enlarged spiny dendritic branches. (**d**) abGCs spine density in groups fed with a control diet was significantly higher than in the PLX5622 diet group (t(38)=6.9, *p=2.98e-08, two-sample t-test). Each dot represents the spine density of an individual GC. The mean of each group is sown by a horizontal line. n = 20 cells from four mice for each of the two groups. Overall, approximately 4000 spines were analyzed. (**e**) A representative high resolution projection image of a dendritic segment with five representative spine heads, marked by an arrowhead along with their measured sizes (in arbitrary units (AU)) in a control mouse, and (**f**) a PLX5622-treated mouse. Scale bar: 10 μm. (**g**) An example of the analyzed dendritic segment and the analysis of spine size in a control mouse and (**h**) a PLX5622-treated mouse. Color intensity reflects the fluorescence intensity. The regions marked for analysis are: the spine head (middle), the adjacent background (left), and the adjacent dendritic shaft (right) (see Methods section for a detailed explanation of the analysis). (**i**) Distributions of abGCs spine sizes in control and PLX5622-treated mice. n = 720 spines from 13 cells from four mice from each group. Two sample Kolmogorov–Smirnov test for probability distributions revealed no group differences (Dn,n'=0.071, p=0.978).
DOI: https://doi.org/10.7554/eLife.30809.004

The following figure supplements are available for figure 2:

**Figure supplement 1.** PLX5622-treated and control mice display a similar global neurogenesis process.
DOI: https://doi.org/10.7554/eLife.30809.005

**Figure supplement 2.** PLX5622-treated and control mice display a similar number of young new-born neurons in the OB.
DOI: https://doi.org/10.7554/eLife.30809.006

**Figure supplement 3.** PLX5622-treated and control mice display similar survival rate of 28-day-old new-born neurons in the OB.
DOI: https://doi.org/10.7554/eLife.30809.007

injected 3 days before the initiation of the diet treatment (i.e., at that time the proliferation of new-born cells was comparable between the groups), the lack of a difference in the number of BrdU-NeuN labeled neurons signifies similar survival rates of the abGCs in the two groups. However, spine density was 25% lower in PLX5622-treated mice as compared with control mice (p<0.001) (*Figure 2b–d*). The morphology of abGCs spines was structurally heterogeneous, with some spines having long necks, some having shorter necks and others with a 'stubby' morphology (i.e., no visible neck). Furthermore, some spiny protrusions ended with only one spine head while others had multiple heads (*Figure 2e–f*). To analyze the spine morphology we measured the spine head size (*Figure 2g–h*), as proxy of synaptic strength (*Matsuzaki et al., 2004*). We found no significant differences between the distributions of spine head size between the PLX5622-treated and control groups (*Figure 2i*).

To examine whether the effects of microglia depletion on spine density are developmentally regulated we conducted an additional experiment assessing the effects of microglial depletion on mature GCs. In this experiment, we injected tdTomato-encoding AAV1 into the RMS, treating the animals with the PLX5622 diet for 25 days, from 45 to 70 d.p.i., that is, when the GCs were already mature (*Figure 3a*). This treatment reduced the numbers of microglia dramatically (*Figure 3—figure supplement 1*), similarly to the effect for animals treated with PLX5622 at 3 to 28 d.p.i. However, following the delayed microglial depletion induction there was no difference between the spine density in the PLX5622-treated and the control groups (*Figure 3b–d*). These results argue that microglia are important for the development of normal synaptic numbers or their maintenance. However, once developed, synapses can mature normally without microglia, as spines show the full breadth of morphologies in microglia-depleted mice.

## Microglia depletion reduces spine dynamics in adult-born neurons

One possible mechanism of controlling spine numbers is differential dynamics of formation and elimination. To assess the role of microglia in abGCs spine dynamics, we used in vivo 2-photon time-lapse imaging. Microglia-depleted mice and their respective controls were infected by a virus injection into the RMS and a window was implanted over the OB (*Livneh and Mizrahi, 2012*). PLX5622 was continuously administered via the diet, from 3 to 28 d.p.i, and spine turnover was assessed by live imaging of spines on both 27 and 28 d.p.i. (*Figure 4a*). By 27 d.p.i., the dendrites of abGCs already reach a stable state (*Livneh and Mizrahi, 2011*), yet they are still in the process of synaptic maintenance, displaying considerable spine dynamics (*Mizrahi, 2007*). Based on the comparison between the two imaging sessions, each spine was denoted as 'stable', 'lost' or 'new' (*Figure 4b,c*; green, red and blue arrows, respectively). In control mice only 70% of the spines remained stable over the 24 hr interval, as compared with 81% of the spines in PLX5622-treated mice (*Figure 4d*) (p<0.001). The percentages of lost and new spines were significantly higher (p<0.005 for each

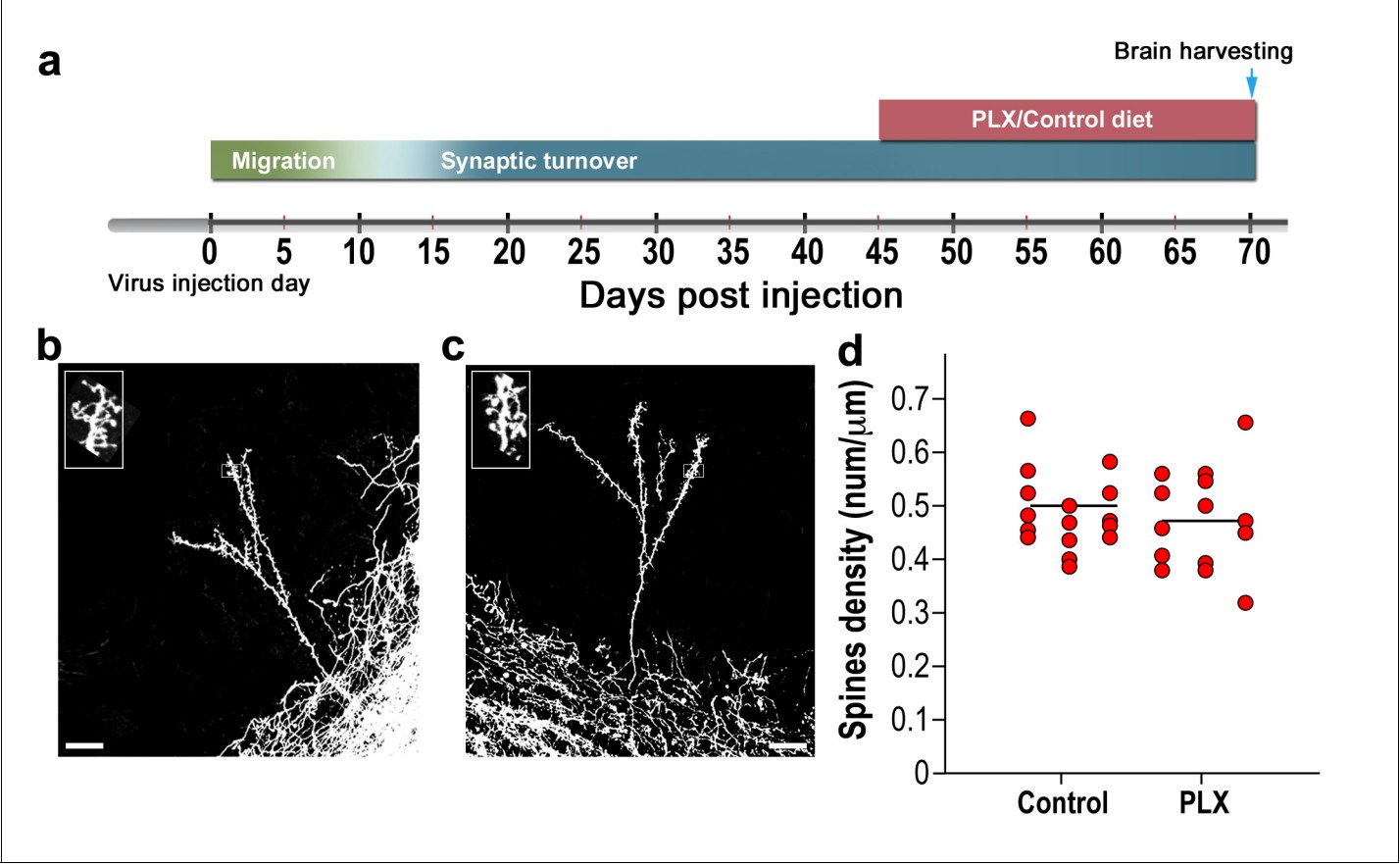

**Figure 3.** Spine density in mature abGCs is not affected by microglia depletion. (**a**) Schematic time-line of the experiment. Adult-born cells were transduced by an injection of Td-Tomato-expressing AAV1 into the RMS. PLX5622/control diet feeding commenced 45 days later, i.e., when the cells were fully mature, and was continued for 25 days. Brains were harvested 70 days after the AAV1 injection. (**b**) High resolution projection images of mature abGCs and their spiny dendritic branches from a control mouse and (**c**) a microglia-depleted (PLX5622-treated) mouse. Scale bar: 20 µm. Insets: enlarged spiny dendritic branches. (**d**) The spine density in mature GCs in mice fed with control diet was similar to the spine density in mice treated with PLX5622 diet (t(34)=0.82, p=0.41, two-sample t-test). Each dot represents the spine density of an individual abGC. The mean of each group is depicted by a horizontal line. n = 18 mature abGCs from four mice for each of the two groups. Overall, approximately 3600 spines were analyzed.
DOI: https://doi.org/10.7554/eLife.30809.008

The following figure supplement is available for figure 3:

**Figure supplement 1.** Delayed PLX5622 treatment (from 45 to 70 d.p.i) induces microglia depletion in the OB.
DOI: https://doi.org/10.7554/eLife.30809.009

comparison) in the control mice as compared with the PLX5622-treated mice (*Figure 4d*). We also calculated the total spine turnover rate [TOR = (number (N)Lost +Nnew) / (N total first session+ N total second session)], finding a relatively high TOR in control mice (0.33 ± 0.01) and lower TOR in PLX5622-treated mice (0.21 ± 0.01; *Figure 4e*) (p<0.001).

Together these findings suggest that microglia have marked effects on the overall synaptic turnover, which in abGCs is very dynamic, underlying the high neuroplasticity potential of these cells. Although the present findings indicate that microglia are involved in both spine formation and elimination, the overall spine density was found to be reduced in the microglia-depleted mice. This may be explained by the fact that during the first 28 days of their maturation abGCs exhibit incrementally increasing spine numbers (i.e., greater spine formation than elimination) (*Whitman and Greer, 2007*; *Livneh and Mizrahi, 2011*), and therefore in microglia-depleted mice the reduction in spine formation during this period is indeed expected to produce an overall reduction in spine density.

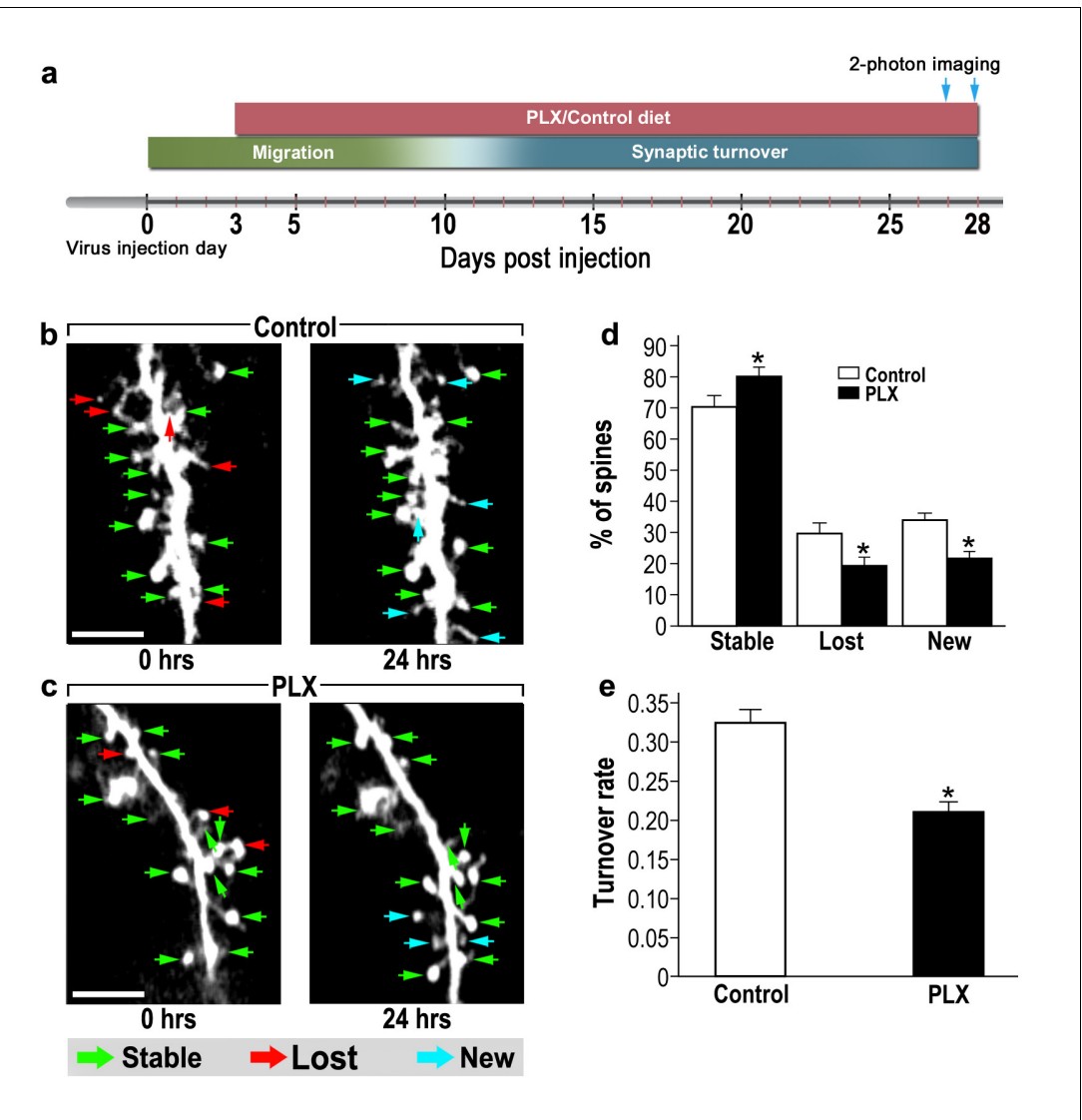

**Figure 4.** abGCs in microglia-depleted mice have lower spine formation and elimination compared with control mice. (a) Schematic time-line of the experiment. Adult-born cells were transduced by administration of Td-Tomato-expressing AAV1 into the RMS. PLX5622/control diet feeding commenced 3 days later. Time-lapse two-photon imaging was performed on days 27 and 28 after the injection. (b) Two projection images of the same abGC dendritic segment, imaged in vivo at a 24 hr interval in a control mouse, and (c) a microglia-depleted (PLX5622-treated) mouse. The green, red and blue arrowheads mark stable, lost and new spines, respectively. Scale bars: 10 µm. (d) Analysis of the in vivo spine dynamics over the 24 hr time lapse (n = 17 dendritic segments from 14 cells in four control mice (for a total of 560 spines), n = 17 dendritic segments from 15 cells in 4 PLX5622-treated mice (for a total of 440 spines)). ANOVA with the group as a between subjects factor and the spine category (Stable, Lost, New) as a within-subjects repeated measure factor revealed an overall group by category interaction ($F(2,64)=18.1$, *$p=7.73e-07$). Specific comparisons using two sample t-tests with Bonferroni's correction revealed significant differences between the control group and the PLX5622-treated group within each category (Stable: $t(32)=3.3$, *$p=0.0018$; Lost: $t(32)=3.38$,*$p=0.0023$; New: $t(32)=5.76$, *$p=2.1-e06$). (e) Mean turnover rate (TOR) of abGCs spines over the 24 hr interval. A significant difference was found between the two groups ($t(32)=6.78$), *$p=1.15e-07$), two sample t-test. Data are presented as the mean ±SEM.
DOI: https://doi.org/10.7554/eLife.30809.010

## The functional implications of microglia depletion

We next tested the impact of a one month long microglia depletion on the odor response profile of the main output neurons of the OB – the mitral cells (MC), which are normally inhibited by abGCs, using in vivo time-lapse two-photon calcium imaging before and after PLX5622 treatment (*Figure 5a*). We implanted three thy1-GCamp3 mice with cranial windows to allow optical access to the MC somata (*Figure 5b*). After recovery (denoted as 'Day 0') and under anesthesia, we imaged calcium responses of MCs to eight monomolecular odors. MC activity was heterogeneous showing a wide range of response profiles (*Figure 5c*); a result consistent with previous reports (*Adam et al., 2014*). These same mice were then depleted from microglia (by treatment with the PLX5622-containing diet for 28 days), after which they were reimaged via the same cranial window, identifying either the same or different cells (*Figure 5a, d and i*). Overall, MC activity at this time point was qualitatively similar to the baseline condition as response profiles were similarly heterogeneous (*Figure 5e*). In fact, the number of odors (out of 8) that MCs responded to was not significantly different before and after microglia depletion (*Figure 5f*). However, a more detailed analysis of the response properties of single neurons showed a significant increase in response magnitude in the microglia-depleted mice (*Figure 5g–h*) (p<0.001). Moreover, a subset of the exact same neurons could be recovered from both imaging sessions, allowing us to analyze the same cells twice, 28 days apart (*Figure 5i–j*). Using this dataset, we plotted the responses of each cell-odor pair before and after microglia depletion (*Figure 5k*). As expected from the population analysis, the distribution of data points was skewed towards increased responsiveness in the microglia depletion condition. A similar analysis using neurons rather than cell-odor pairs also showed a significant increase in neuronal responses magnitude following microglia depletion (*Figure 5l*) (p<0.0001). These results show that the general response profile of MCs was maintained. However, a significant strengthening of mean MC responsiveness (of about 29%) was evident.

## Microglial *Cx3cr1* deficiency reduces spine number and size in adult-born neurons

To influence the formation and elimination of dendritic spines during the adult neurogenesis process microglia have to interact with the developing new-born neurons. One important mode of neuronal-microglia communication involves signaling via the CX3CR1 (*Paolicelli et al., 2014*; *Pagani et al., 2015*) which in the brain is expressed almost exclusively by microglia (*Jung et al., 2000*). To elucidate the importance of CX3CR1-mediated microglial-neuronal interactions for the development and maintenance of dendritic spines in adult new-born neurons we used mice with a genetic deletion of *Cx3cr1* (*Jung et al., 2000*).

We carried out experiments similar to those described above (*Figure 6a*), except that the density and size of dendritic spines on abGCs were compared between $Cx3cr1^{-/-}$ and wild-type (WT) control mice. The overall neurogenesis process in the $Cx3cr1^{-/-}$ mice seemed to be similar to that of WT mice (*Figure 6—figure supplement 1a–b*). The number of dendrites per cell (*Figure 6—figure supplement 1c*) and average length of these dendrites (*Figure 6—figure supplement 1d*) were comparable. The number of young new-born neurons (labeled by DCX) was also comparable between the $Cx3cr1^{-/-}$ animals and controls (*Figure 6—figure supplement 2a–d*). However, spine density was 31% lower in $Cx3cr1^{-/-}$ mice as compared with WT mice (p<0.001) (*Figure 6b–d*). The morphology of abGCs spines was structurally heterogeneous (*Figure 6e,f*). However, in contrast to the absence of an effect following microglia depletion, here there was a significant difference between the distribution of spine head size between $Cx3cr1^{-/-}$ and WT mice (p<0.01) (*Figure 6g*). Specifically, the distribution was skewed towards smaller spine head sizes in $Cx3cr1^{-/-}$ mice, such that the frequency of small spine heads (up to 20 AU) was significantly higher in $Cx3cr1^{-/-}$ than in WT mice (p<0.01), whereas the frequency of big spine heads (50–60 AU) was significantly lower in $Cx3cr1^{-/-}$ than in WT mice (p<0.01).

These findings show that impaired neuronal-microglial interactions due to *Cx3cr1* deficiency induces a marked reduction in spine density and size, suggesting that CX3CR1 signaling may indeed be a mechanism underlying the effects of microglia on spine development and maturation in abGCs.

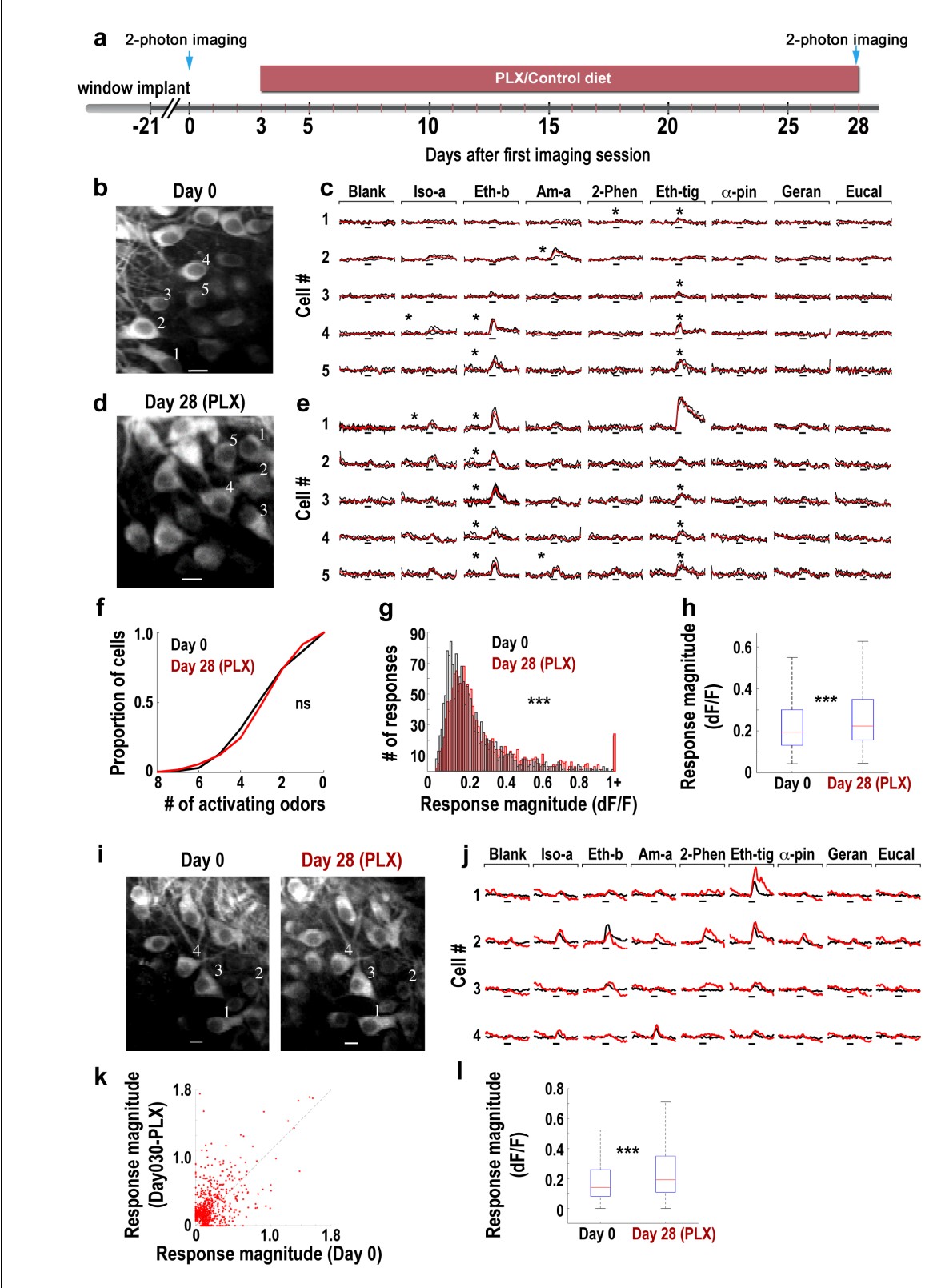

**Figure 5.** Response magnitude of MCs increases following microglia depletion. (a) Schematic time-line of the experiment. Three Thy1-GCamp3 mice with implanted cranial windows were imaged for calcium responses of MCs to eight monomolecular odors, both at baseline (day 0) and 30 days later, following microglia depletion induced by PLX5622 treatment. (b) In vivo two-photon micrograph of a representative field of MCs at the first imaging session (day 0). Scale bar = 10 µm. (c) Examples of calcium transients from the five neurons marked on the left image in response to eight odors during

*Figure 5 continued on next page*

*Figure 5 continued*

baseline. Odor stimulation was applied for 2 s, denoted as a black line under each trace. Black traces represent the three single trials and the red trace is the average. Asterisks denote a statistically significant response to that odor. Scale - 100% ΔF/F. Iso-a = Isoamyl acetate, Eth-b = Ethyl butyrate, Am-a = Amyl Acetate, 2-Phen = 2 phenylethanol, Eth-tig = Ethyl tiglate, α-pin = Alpha Pinen, Geran = Geraniol, Eucal = Eucalypyol. (d) Same as in panel b but for data collected from a different mouse, following 30 days of PLX5622 administration. (e) Examples of calcium transients from five neurons marked on the left image in response to eight odors. (f) Cumulative distribution of the proportion of MCs responding to 0–8 odors, before (Day 0; n = 622 MCs) and after (Day 30; n = 575 MCs) microglia depletion. The distributions in the two groups were not significantly different (Dn,n'=0.0649, p=0.155, two-sample Kolmogorov-Smirnov test). (g) Histograms showing the response magnitudes of all the significant cell odor pair responses. Data is based on imaging 622 neurons for a total of n = 1641 odor pairs on Day 0 (black bars), and 575 neurons for a total of n = 1492 odor pairs on Day 30 (red bars). Distributions are significantly different (Dn,n'=0.1114, ***p=6.30e-09, two-sample Kolmogorov–Smirnov test). (h) Box plots for the distributions shown in (g) (t(3131) = −4.9771, ***p=6.8e-07, two sample t-test). (i) In vivo two-photon micrographs of the same field of MCs before and after microglia depletion. Scale bar: 10 μm. (j) Representative examples of the mean calcium transients evoked by eight odors before (black) and after (red) microglia depletion. Data is from the four neurons marked on the image in F. Scale - 100% ΔF/F. Odors are the same as in A. (k) Scatter plot of the individual cell-odor pair responses from the neurons at day 0 (x-axis) and at day 30 (y-axis). n = 180 neurons with a total of n = 654 cell-odor pairs. (l) Box plots of the repeated neuronal responses before and after microglia depletion (t(653)=6.3525, ***p=3.97e-10, paired-t-test).

DOI: https://doi.org/10.7554/eLife.30809.011

## Microglial *Cx3cr1* deficiency reduces spine dynamics in adult-born neurons

To assess the role of CX3CR1 signaling in abGCs spine dynamics, we used in vivo 2-photon time-lapse imaging. An experimental design similar to the one employed in the microglia-depletion experiment was used; *Cx3cr1*$^{-/-}$ and WT control mice were compared (*Figure 7a*). Based on the comparison between the two imaging sessions each spine was denoted as 'stable', 'lost' or 'new' (*Figure 7b,c*; green, red and blue arrows, respectively).

In WT mice only 70% of the spines remained stable over the 24 hr interval, as compared with 85% of the spines in *Cx3cr1*$^{-/-}$ mice (*Figure 7d*) (p<0.001). The percentages of lost and new spines were significantly higher (p<0.005 for each comparison) in the WT mice as compared with the *Cx3cr1*$^{-/-}$ mice (*Figure 7d*). Total spine turnover rate in WT mice was higher (0.33 ± 0.01) as compared to the *Cx3cr1*$^{-/-}$ mice (0.16 ± 0.01; *Figure 7e*) (p<0.001). Together, these findings show that the effects of impaired neuronal-microglial interactions due to *Cx3cr1* deficiency recapitulate the findings with microglial depletion, suggesting that CX3CR1 signaling is an important mechanism for microglial involvement in synaptic turnover.

## Microglial *Cx3cr1* deficiency impairs the normal interactions between microglia and adult-born neurons

The lower densities and sizes of abGCs spines in *Cx3cr1*$^{-/-}$ mice could result from an altered mode of interaction between the *Cx3cr1*-deficient microglia and the abGCs. To test this hypothesis, we compared the number of putative contacts between microglial processes and abGCs' dendrites in *Cx3cr1*$^{-/-}$ vs. WT mice (*Figure 8a–e* ; *Videos 1*, *2*). The number of putative contacts between microglia and dendritic shafts was 38% lower in *Cx3cr1*$^{-/-}$ mice compared to WT mice (p<0.001) (*Figure 8f*). In contrast, there were more interaction between microglia and spines in *Cx3cr1*$^{-/-}$ mice than in WT mice. Specifically, while putative microglia-spine contacts were detected on 9.0% of all spines in WT mice, in *Cx3cr1*$^{-/-}$ the frequency of such contacts was 12.1% (*Figure 8g*) (p<0.05). The altered frequency of microglia-spine/dendritic shaft interaction in *Cx3cr1*$^{-/-}$ mice was probably not a result of differential microglial morphology, because the number, length, and spatial dispersion of microglial processes was comparable in *Cx3cr1*$^{-/-}$ and WT mice (*Figure 8—figure supplement 1*). Moreover, the average densities of microglia in the OB were also comparable between the groups (*Figure 8—figure supplement 2*)

## Comparison of the cytokine/chemokine expression profile between the two microglia manipulation models

In the studies described above we examined the effects of two manipulations - microglia depletion and *Cx3cr1* deletion. Although the findings obtained in these models are similar, it should be acknowledged that the two models may be associated with distinct micro-environments in the brain. In particular, the CX3CR1 is a receptor through which neurons suppress microglial activity

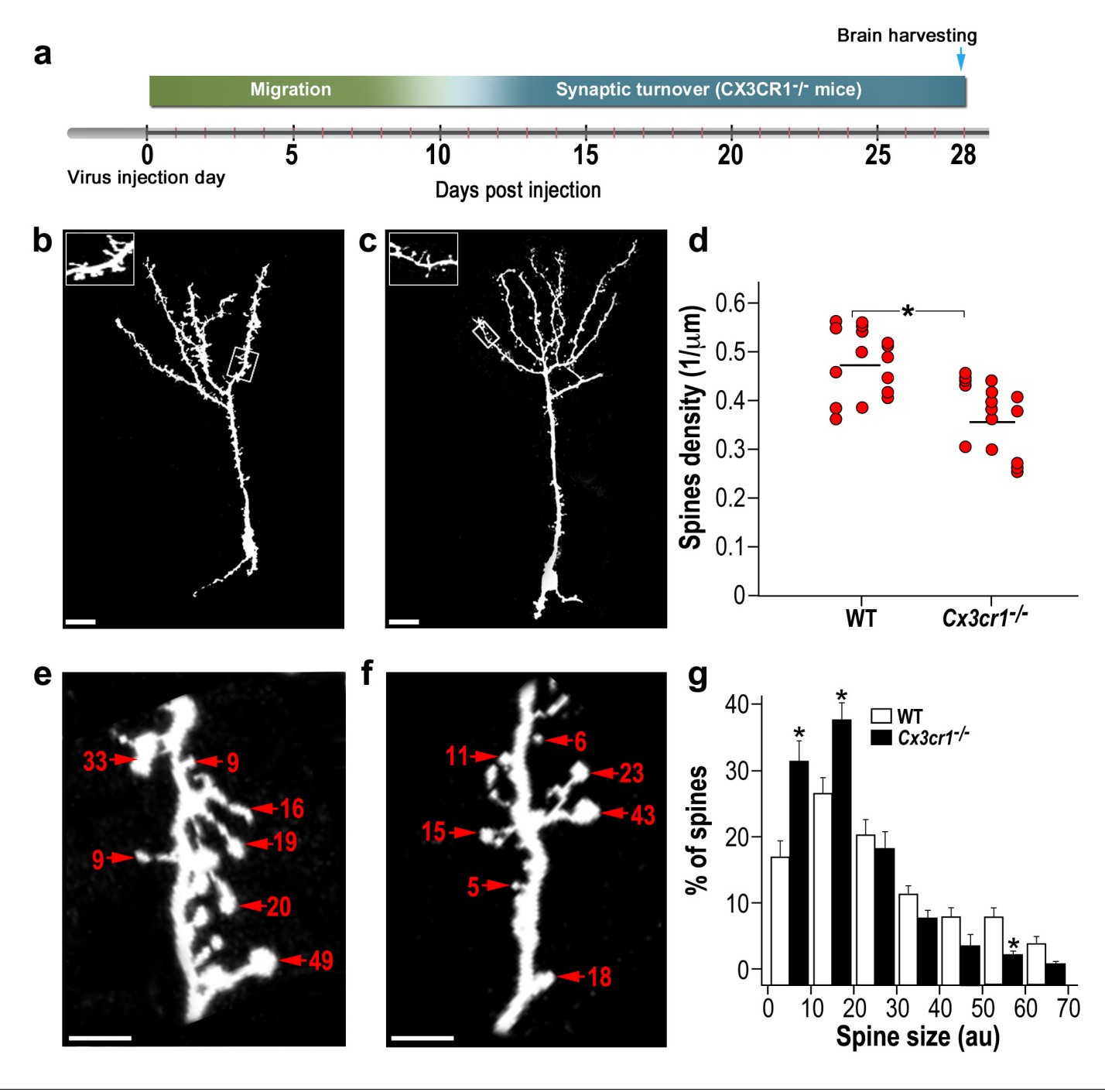

**Figure 6.** abGCs in *Cx3cr1⁻ᐟ⁻* mice have lower spine density and smaller spine sizes compared to WT mice. (**a**) Schematic time-line of the experiment. Adult-born cells were transduced with a TdTomato-expressing AAV1 injection into the RMS. (**b**) High resolution projection images of abGCs and their spiny dendritic branches from a WT control mouse, and (**c**) a *Cx3cr1⁻ᐟ⁻* mouse. Scale bar: 20 µm. Insets: enlarged spiny dendritic branches. (**d**) abGCs spine density in the WT group was significantly higher than in the *Cx3cr1⁻ᐟ⁻* group (t(38)=5.1; *p=9.71e-06, two-sample t-test). Each dot represents the spine density of an individual GC. The mean of each group is shown by a horizontal line. n = 20 cells from five mice for each of the two groups. Overall, approximately 4000 spines were analyzed. (**e**) Representative high resolution projection images of dendritic segments from a WT and (**f**) a *Cx3cr1⁻ᐟ⁻* mouse, with seven representative spine heads, marked with arrowheads along with their measured sizes (in arbitrary units (AU). Scale bar: 10 µm. (**g**) The overall distributions of abGCs spine sizes (based on n = 760 spines from 13 and 14 cells from four mice from each group) was significantly different between WT and *Cx3cr1⁻ᐟ⁻* mice (Dn,n'=0.237; p=0.0078 in the Kolmogorov–Smirnov test for probability distributions), and specifically for spine sizes 0–10 au (t(25)=3.91, *p=6.2e-04), 10–20 au (t(25)=3.14, *p=0.004), and 50–60 au (t(25)=3.69, *p=0.001), two-sample t-tests (with Bonferroni's correction).
*Figure 6 continued on next page*

*Figure 6 continued*

DOI: https://doi.org/10.7554/eLife.30809.012

The following figure supplements are available for figure 6:

**Figure supplement 1.** *Cx3cr1*$^{-/-}$ and WT mice display a similar global neurogenesis process.

DOI: https://doi.org/10.7554/eLife.30809.013

**Figure supplement 2.** *Cx3cr1*$^{-/-}$ and control mice display similar young new-born neurons number in the OB.

DOI: https://doi.org/10.7554/eLife.30809.014

(*Cardona et al., 2006*; *Wolf et al., 2013*). Therefore, in *Cx3cr1*$^{-/-}$ mice the micro-environment may be pro-inflammatory and influencing neurogenesis via mechanisms different than those underlying the effects of microglia depletion. To compare the inflammatory environment in the two models we examined the profile of cytokine and chemokine expression levels in the OB of PLX5622-terated, *Cx3cr1*$^{-/-}$ and their respective control mice. An RNA-seq analysis of OBs isolated from these groups revealed no significant differences in the expression of most of the major inflammatory cytokines and chemokines (*Supplementary file 2*). Out of the 40 cytokine/chemokines measured, only two molecules exhibited differential expression, including a lower *Il-16* (a microglial enriched gene) expression in the PLX5622-treated group as compared to the control group, and lower *Ccl5* expression in *Cx3cr1*$^{-/-}$ mice as compared to the PLX5622-treated mice.

To further compare the two models, we also analyzed the protein levels of the same 40 cytokines and chemokines, using a mouse cytokine array. The levels of most of the inflammatory molecules within the OB (including IL-16 and CCL5) were very low (below the detection limit of this assay). The levels of the molecules that were above the detection limit were lower in the PLX5622-treated and *Cx3cr1*$^{-/-}$ mice compared with the levels in WT mice, but these differences did not reach statistical significance (*Figure 9a* and *Figure 9—source data 1*). To examine a possible peripheral contribution to the inflammatory status in the two models we examined the levels of serum inflammatory cytokines and chemokines in PLX5622-treated, *Cx3cr1*$^{-/-}$ and WT mice. Again, the levels of most of these molecules were below the detection limit of the assay. Among the molecules whose levels were above the detection limit, four molecules were different (usually reduced) in one of the models compared to the WT control group, but there were no significant differences between the levels of these molecules in PLX5622-treated compared to *Cx3cr1*$^{-/-}$ mice (*Figure 9b* and *Figure 9—source data 1*).

## Discussion

The findings of this study demonstrate that microglia play an important role in the initial development, maintenance, plasticity and physiological impact of synapses on adult-born neurons. Furthermore, microglia-neuron communication through CX3CR1 signaling serves as an important mechanism underlying the role of microglia in these processes (*Figure 10*). In mature abGCs the microglial involvement in synaptic regulation (under basal/quiescent conditions) subsides. The results importantly extend previous research on the global role of microglia in regulating adult-born neuron numbers under various physiological and pathological conditions (*Ekdahl et al., 2003*; *Monje et al., 2003*; *Butovsky et al., 2006*; *Ziv et al., 2006*; *Sierra et al., 2010*; *Maggi et al., 2011*; *Reshef et al., 2014*), by elucidating the specific roles of microglia and their CX3CR1 signaling in the adult neurogenesis process.

### Effects of microglia depletion on synaptic development of abGCs

Most previous studies on the role of microglia in synaptic regulation focused on the early postnatal brain developmental period. During this period, microglia play an important modulatory role in the development of global neuronal circuit connectivity (*Wu et al., 2015*). This modulation involves the pruning of synapses by microglia, based on two distinct microglial molecular pathways, as evidenced by reduced pruning and altered networks connectivity in developing mice with either microglial *Cx3cr1* deficiency or complement receptor 3 (CR3) deficiency (*Paolicelli et al., 2011*; *Schafer et al., 2012*; *Zhan et al., 2014*). Moreover, in sensory-deprived adult animals, microglia were found to be involved in pruning of spines on OB abGCs (*Denizet et al., 2017*). We report here, that microglia-

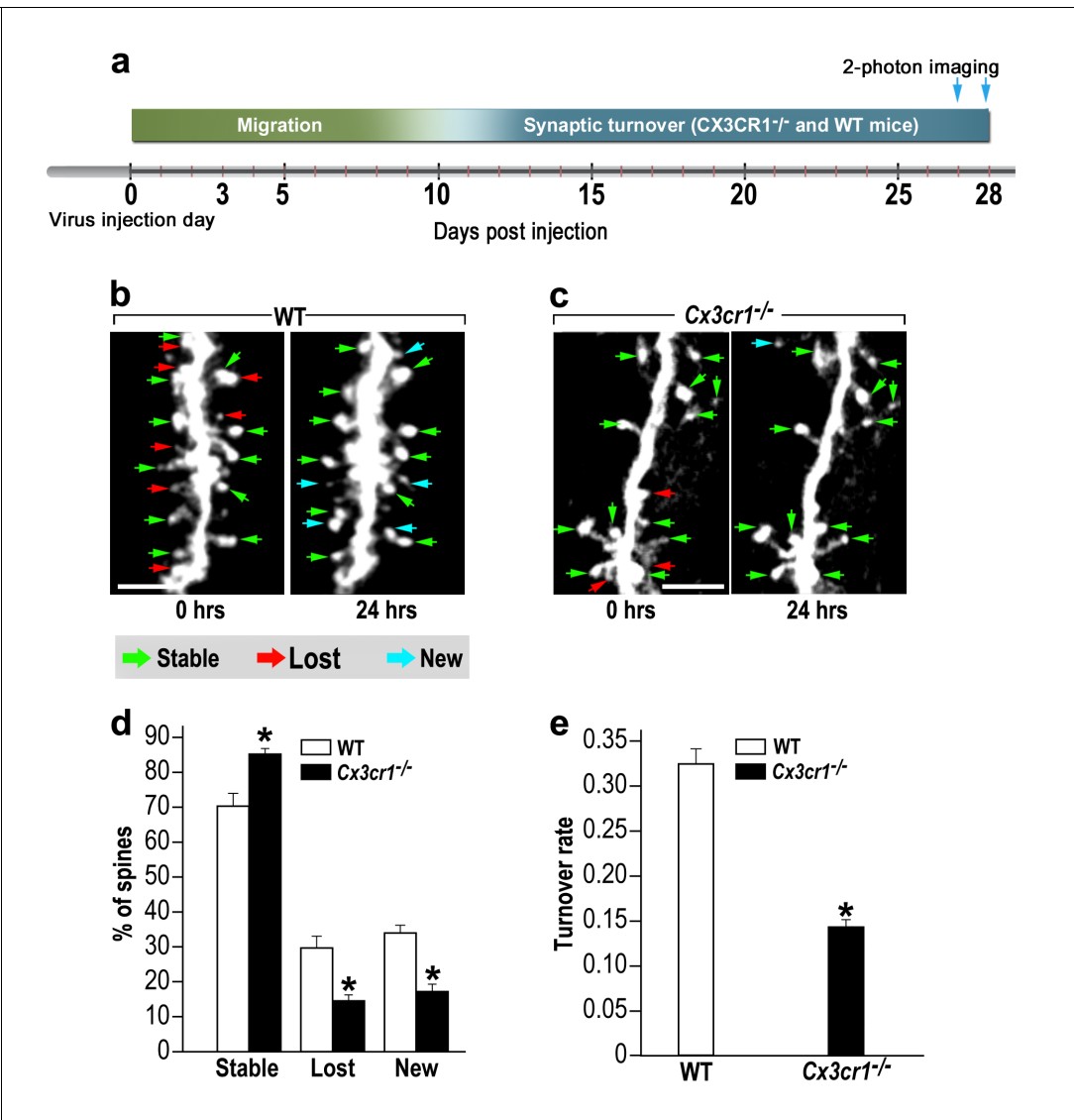

**Figure 7.** abGCs in *Cx3cr1*[-/-] mice have lower spine formation and elimination compared with WT mice. (**a**) Schematic time-line of the experiment. Adult-born cells were transduced with a TdTomato-expressing AAV1 injection into the RMS. (**b**) Two projection images of the same abGC dendritic segment are depicted, imaged in vivo at a 24 hr interval in a WT mouse, and (**c**) a *Cx3cr1*[-/-] mouse. The green, red and blue arrowheads mark stable, lost and new spines, respectively. Scale bars: 10 μm. (**d**) Analysis of the in vivo spine dynamics over the 24 hr time lapse (n = 16 segments from 14 cells in 4 WT mice, n = 538 spines; n = 18 segments from 16 cells in 4 *Cx3cr1*[-/-] mice n = 450 spines). ANOVA with the group as a between subjects factor and the spine category (Stable, Lost, New) as a within-subjects repeated-measures factor revealed an overall group by category interaction (F(2,64)= 48.1, p=1.77e-13). Specific comparisons using two sample *t*-tests with Bonferroni's correction revealed significant differences between the WT group and the *Cx3cr1*[-/-] group within each category (Stable: t(32)=6.0, *p=1.08e-06; Lost: t(32)=6.1, *p=8.1e-071; New: t(32)=8.7,*p=6.9e-10), two sample t-test. (**e**) Mean turnover rate (TOR) of abGCs spines over the 24 hr interval. A significant difference was found between the two groups (t(32)=9.6; *p=6.03e-11). Data presented as the mean ±S.E.M.

DOI: https://doi.org/10.7554/eLife.30809.015

depleted mice displayed reduced spine elimination under normal physiological conditions. Thus, similarly to their role in early brain development, microglia are also involved in pruning of dendritic spines in normally-developing adult-born neurons. Surprisingly, microglia depletion was associated with reduced spine density on mature abGCs. This result is probably related to the present finding that in addition to the reduction in spine elimination, microglia depleted mice also exhibited reduced formation of spines on abGCs. This finding is consistent with a previous report, showing that microglia depletion (using a genetic ablation method) reduced plasticity-related spine formation

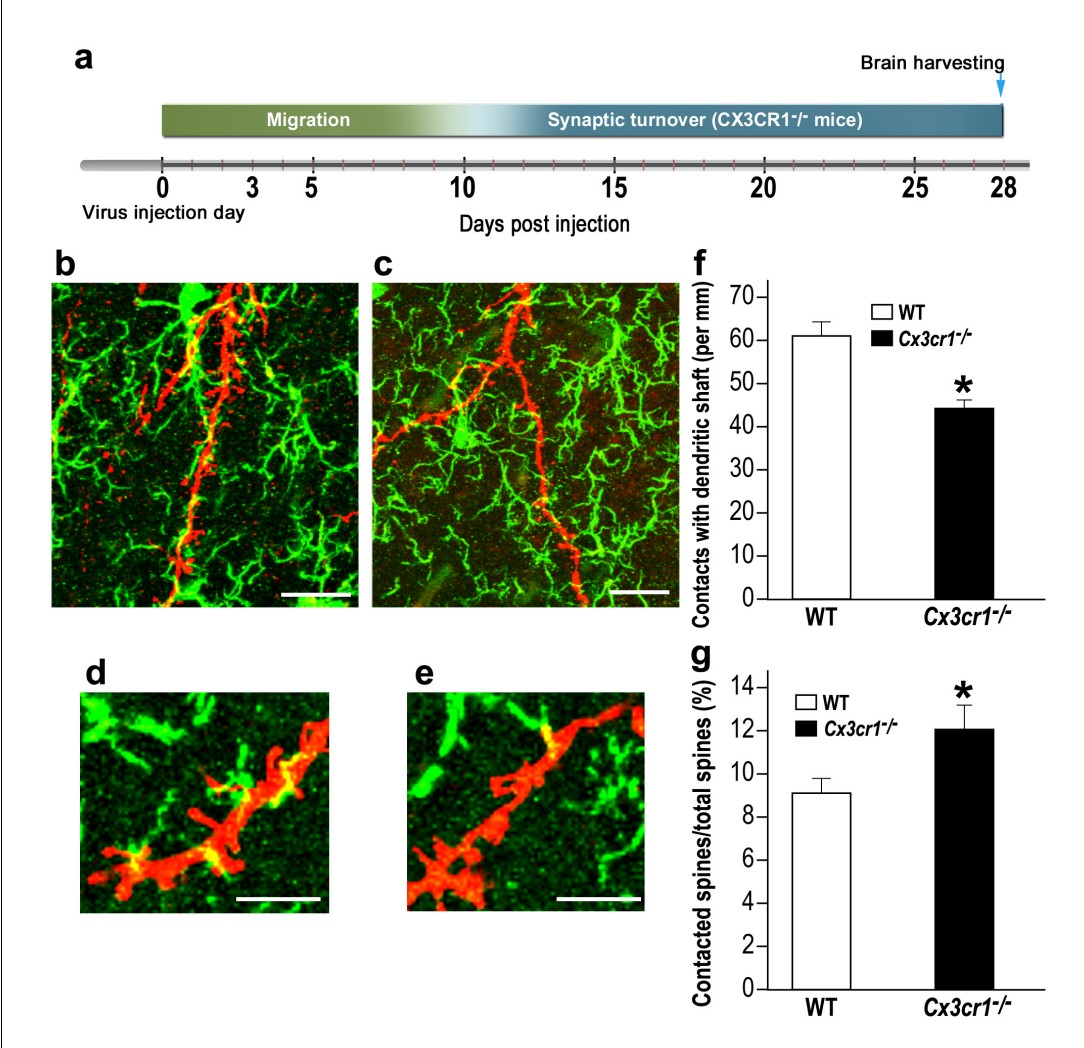

**Figure 8.** *Cx3cr1* deficient microglia show aberrant interactions with abGCs. (**a**) Schematic time-line of the experiment. Adult-born cells were transduced with a TdTomato-expressing AAV1 injection into the RMS, and the pattern of microglia-dendrites contacts were analyzed 28 days later. (**b**) High resolution projection images showing the interaction between microglia and an abGC dendrite from a WT mouse, and (**c**) a *Cx3cr1*<sup>−/−</sup> mouse (abGC dendrite = red; microglia = green). Scale bar: 20 μm. (**d**) High magnification of the interaction between microglial processes and a single dendritic segment from a WT mouse, and (**e**) a *Cx3cr1*<sup>−/−</sup> mouse. Scale bar: 10 μm. (**f**) Density of the contacts between microglia and dendritic shaft (expressed as the contact number per mm length of the shaft) in the WT and *Cx3cr1*<sup>−/−</sup> groups. n = 13 cells from four animals in each group. Overall, 70 dendrites were analyzed (t(24)=4.1, *p=4.1e-04, two-sample t-test). (**g**) The percentage of spines contacted by microglia out of the total spine population in the WT and *Cx3cr1*<sup>−/−</sup> groups. n = 13 cells from four animals in each group. Overall, approximately 2400 spines were analyzed (t(24)=2.2, *p=0.03, two-sample t-test).

DOI: https://doi.org/10.7554/eLife.30809.016

The following figure supplements are available for figure 8:

**Figure supplement 1.** Microglia in *Cx3cr1* deficient and WT mice have similar morphology.

DOI: https://doi.org/10.7554/eLife.30809.017

**Figure supplement 2.** *Cx3cr1* deficient and WT mice display similar microglial density in the OB.

DOI: https://doi.org/10.7554/eLife.30809.018

in mature neurons of the adult motor cortex following motor learning (*Parkhurst et al., 2013*). Thus, microglia play a role in both synapse formation and elimination. The decrease in spine density may stem from the spine dynamics of these cells. Specifically, in abGCs there is an initial period of greater spine formation than elimination up to 28 days after their generation, followed by a period of relative stability and a subsequent period of greater pruning from 3 to 6 months (*Livneh and Mizrahi, 2011*). Thus, during the first few weeks of abGCs development, microglia are predominantly

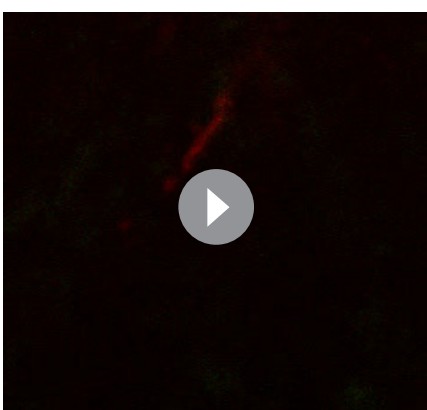

**Video 1.** Putative contacts between microglial processes and abGCs in a WT mouse. A movie demonstrating putative contacts between WT microglia and the dendrites of an abGC. The movie shows the individual focal planes of a Z stack obtained using confocal microscopy. The analysis of the density of microglia-dendritic contacts reported in *Figure 8* was conducted based on examination of such individual focal planes.

DOI: https://doi.org/10.7554/eLife.30809.019

involved in spine formation, and therefore microglia depletion during that period resulted in reduced spine density.

Consistently with the above interpretation, when the PLX5622 treatment was initiated 7 weeks following the birth of the labeled abGCs, when these cells are considered to be mature, the microglia depletion had no effect on spine density. In this experiment, the microglia depletion occurred when spine density is known to be stable (*Livneh and Mizrahi, 2011*), as verified here by showing that spine density in 70-day-old abGCs (*Figure 3d*) was similar to the spine density in 28-day-old abGCs (in control diet-treated animals – *Figure 2d*). Thus, the role of microglia in the regulation of dendritic spines in abGCs (under normal quiescent conditions) is limited to the initial developmental period of these cells (up to 28 days after birth). Interestingly, in late adulthood (i.e., in 10-months-old mice) microglia depletion was found to increase spine density (*Rice et al., 2015*). This finding suggests that along the aging process, when spine density is destabilized and spine loss commences (*Dickstein et al., 2013*), microglia again assume an important regulatory role and mediate age-associated spine pruning.

## Effects of microglial *Cx3cr1* deficiency on synaptic development of abGCs

In *Cx3cr1*$^{-/-}$ mice the neuronal-microglial communication through CX3CR1 is abrogated (*Paolicelli et al., 2014*). Thus, this model provides a more specific tool than global microglia depletion to probe the mechanisms by which microglia affect synaptic development. We report here that *Cx3cr1*$^{-/-}$ mice displayed reduced spine elimination under normal physiological conditions, indicating that similarly to their role in early brain development (*Paolicelli et al., 2014*), microglia communication with neurons via CX3CR1 signaling are also involved in pruning of dendritic spines in normally-developing adult-born neurons. However, in contrast with early development, when *Cx3cr1* deficiency is associated with a transient increase in spine density (*Paolicelli et al., 2011*), *Cx3cr1* deficiency in the adult brain is associated with reduced spine density in developing abGCs. This discrepancy can be explained by the current finding that *Cx3cr1*$^{-/-}$ mice also exhibit a reduction in spine formation in abGCs. These findings suggest that the overall effect of microglia on spine density is different in neonatal- vs. adult-born neurons, based the differential developmental dynamics of spines during these periods.

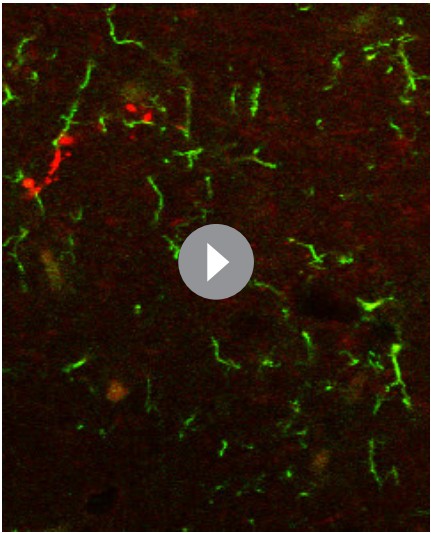

**Video 2.** Putative contacts of *Cx3cr1* deficient microglia processes and abGCs. A movie demonstrating putative contacts between *Cx3cr1*-deficient microglia and the dendrites of an abGC. The movie shows the individual focal planes of a Z stack obtained using confocal microscopy. The analysis of the density of microglia-dendritic contacts reported in *Figure 8* was conducted based on examination of such individual focal planes.

DOI: https://doi.org/10.7554/eLife.30809.020

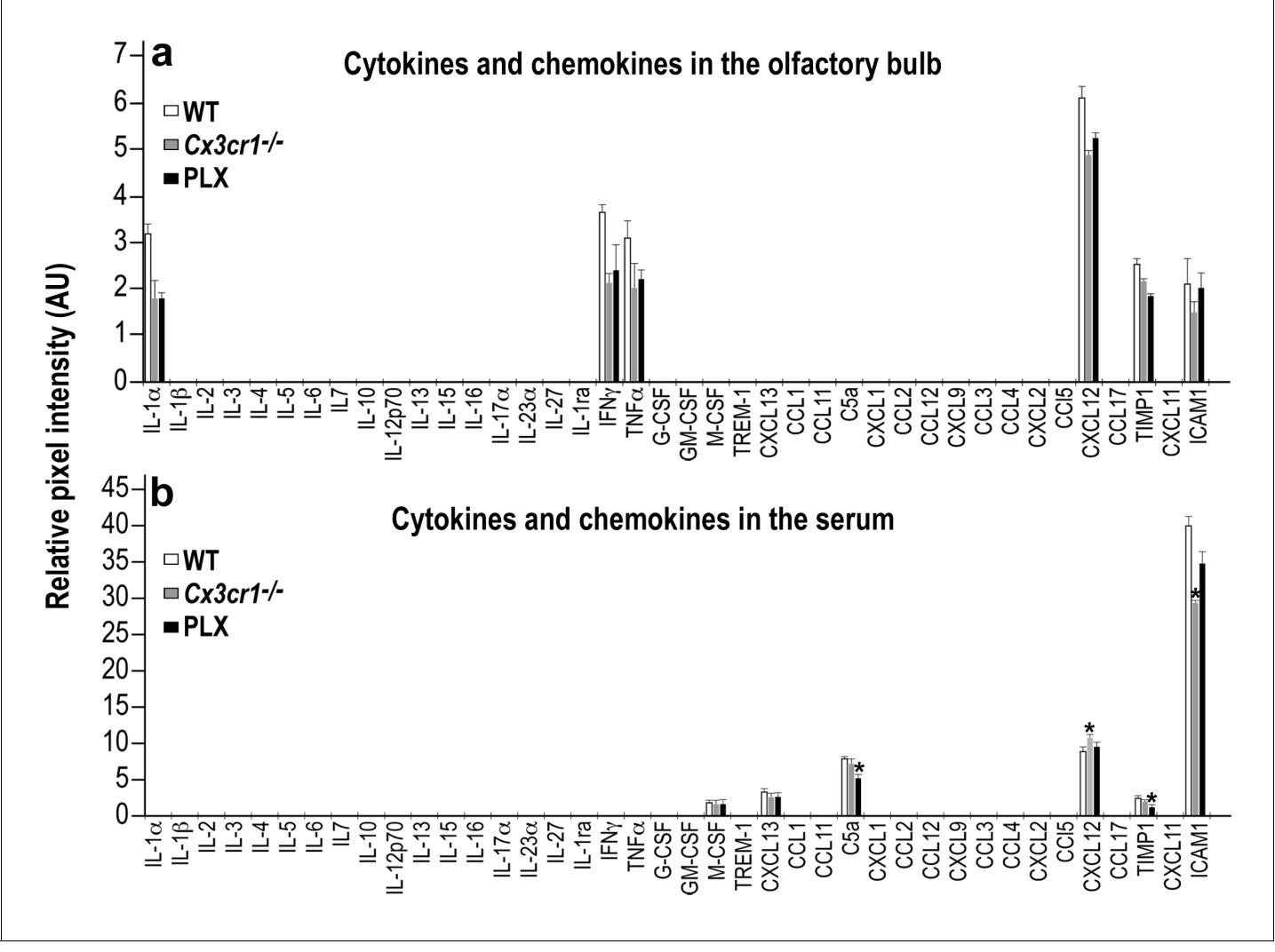

**Figure 9.** Cytokine and Chemokine expression in the OB and serum of WT, *Cx3cr1*$^{-/-}$ and PLX5622-treated mice. (**a**) Cytokine and Chemokine microarray assay performed on total tissue lysates of OBs from WT, *Cx3cr1*$^{-/-}$, and PLX5622-terated mice. Expression levels are presented as arbitrary units, measured by densitometry. Using the Bonferroni procedure to correct for multiple comparisons, no significant changes were found between the expressed cytokines. *n* = 5, pooled in each group. Two-sample *t*-test. (**b**) Cytokine and Chemokine microarray assay performed on serum from WT, *Cx3cr1*$^{-/-}$, and PLX5622-terated mice. Cytokine and Chemokine expression levels were measured by densitometry, and are presented as arbitrary units. Using the Bonferroni procedure to correct for multiple comparisons, significant changes were found in the following molecules: (C5a: WT vs PLX, t(2) =12.1, *p=0.0067; CXCL12: WT vs KO, t(2)=11.7, *p=0.007; TIMP1: WT vs PLX, t(2)=8.1, *p=0.014; ICAM1: WT vs KO, t(2)=10.4, *p=0.009). *n* = 5 pooled in each group. Two-sample *t*-test.

DOI: https://doi.org/10.7554/eLife.30809.021

The following source data is available for figure 9:

**Source data 1.** Data of Cytokine and Chemokine expression in the OB and serum of WT, *Cx3cr1*$^{-/-}$ and PLX5622-treated mice.
DOI: https://doi.org/10.7554/eLife.30809.022

Specifically, in the early postnatal OB, the number of spines increases rapidly over the first 2 weeks, reflecting higher spine formation than elimination, immediately followed by a period of spine reduction, reflective of greater spine pruning (*Brunjes et al., 1982*). Thus, during the 3$^{rd}$ third week of life (the time point at which *Cx3cr1* deficiency was found to be associated with increased spine density [*Paolicelli et al., 2011*]), microglia are predominantly involved in pruning. In contrast, as noted above, during the development of abGCs the higher rates of spine formation than elimination lasts longer (until 28 days following abGCs generation) (*Livneh and Mizrahi, 2011*). During this period,

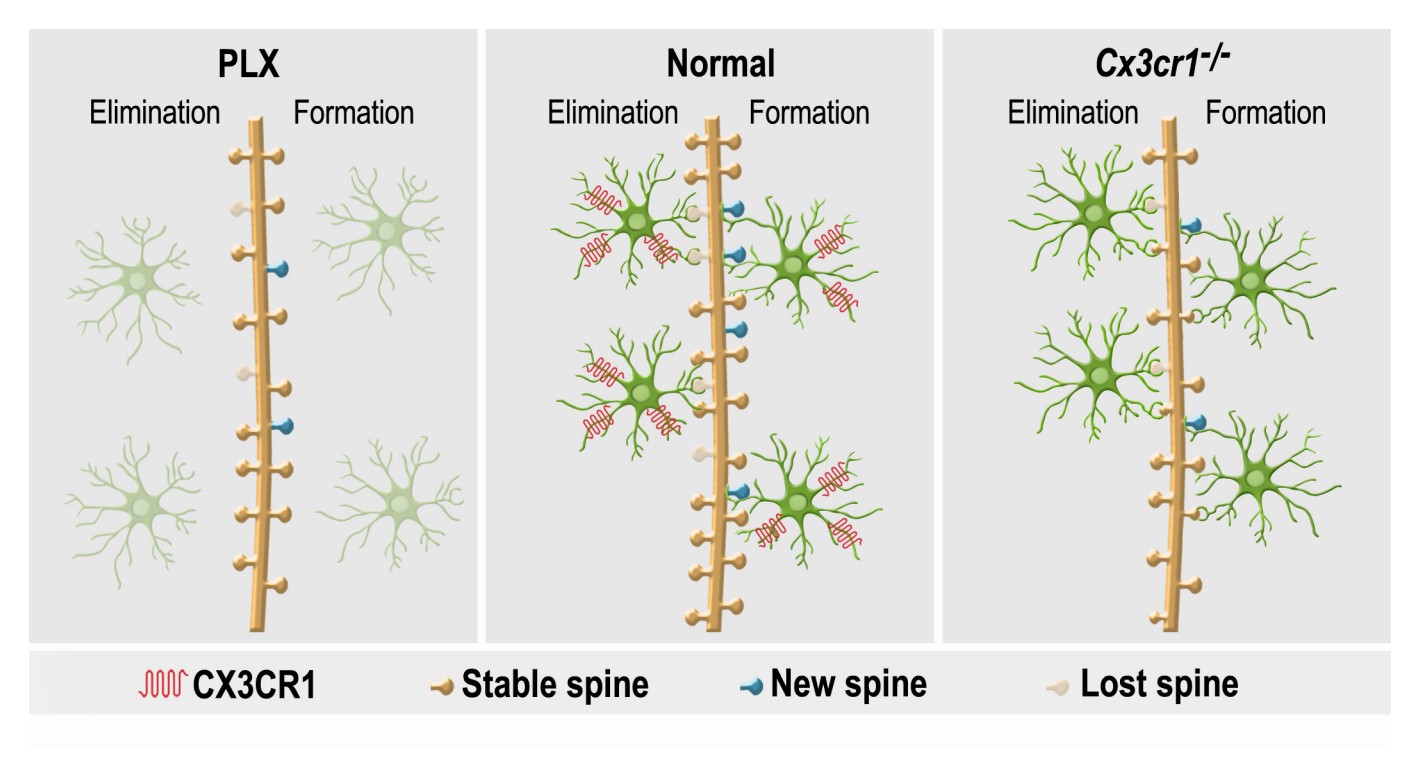

**Figure 10.** A summary of the effects of microglial depletion and *Cx3cr1* deficiency on the formation, elimination, head size and density of spines on abGCs. Under Normal conditions (middle panel), microglia (green cells) play an important role in both facilitation of spine formation and in elimination of nonfunctional spines, leading to normal maturation and plasticity of adult-born granule cells (abGCs) in the OB. These effects are at least partly dependent on the microglial expression of CX3CR1, which mediates neuronal-microglial communication. Microglial depletion in PLX5622-treated mice (left panel) reduced the formation and elimination of spines on abGCs. Because in control mice spine formation exceeds spine elimination during the initial developmental period of abGCs, the reduced synaptic turnover in microglia-depleted mice resulted in lower spine density. Abrogation of normal microglia-neuronal interactions in *Cx3cr1*$^{-/-}$ mice (right panel) induced a reduction in spine formation, possibly due to the lower number of contacts between microglial processes and dendritic shafts, as well as lower spine head size, possibly due to the increased proportion of spines that were contacted by microglia. Spine elimination was also reduced, suggesting that CX3CL1-CX3CR1-mediated communication between specific spines on abGCs and adjacent microglial processes plays an important role in spune pruning. Similarly to the effects of microglia depletion, spine density was also reduced in the *Cx3cr1*$^{-/-}$ mice.

DOI: https://doi.org/10.7554/eLife.30809.023

microglia are predominantly involved in spine formation, and therefore microglial aberrations (e.g., depletion or *Cx3cr1* deficiency) were found here to be associated with reduced spine density.

Our finding that *Cx3cr1*$^{-/-}$ abGCs had smaller spine heads suggests that in addition to their role in spine formation and elimination, microglia modulate spine morphology. Because spine size is positively correlated with synaptic efficacy (*Holtmaat et al., 2005*), it may be suggested that microglia regulate synaptic strength. The finding that microglia depletion did not cause any change in spine morphology, suggests that microglia are not essential for the regulation of spine head size. Rather, the enhanced putative microglia-to-spine contacts in *Cx3cr1*$^{-/-}$ mice may be the mechanism underlying the lower spine head size, given previous direct evidence that contacts between microglia and spines in the developing visual cortex can cause a transient reduction in spine size (*Tremblay et al., 2010*). Our findings extend these results by showing that in abGCs such changes can be persistent and cumulative, resulting in almost doubling of the proportion of small spines out of the total spine population (*Figure 6g*), possibly reflecting a delay in the normal maturation of synapses.

Our results demonstrate that CX3CL1-CX3CR1 signaling is an important mechanism underlying the regulation of synaptic development in abGCs. In the CNS, it has been shown that CX3CL1 induces chemotaxis of microglia cells to location of neuronal damage (*Harrison et al., 1998*; *Fuhrmann et al., 2010*), however, no studies were conducted yet to directly examine the role of this

system in the interactions between microglial and neuronal components under normal quiescent conditions. The present finding, showing that *Cx3cr1*-deficient microglia made less putative contacts with dendritic shafts of abGCs, suggests a role for the CX3CL1-CX3CR1 system in such contacts. Furthermore, the reduction in microglial-dendritic shaft contacts may account for the lowered spine formation and subsequent reduction in spine densities on abGCs, consistently with a recent study showing that during brain development contacts between microglia and dendritic shafts are essential for spine formation (*Miyamoto et al., 2016*). The finding that the mere impairment in CX3CR1 signaling produces effects on spine formation and density that are similar to the effects of global microglial-depletion highlights the importance of the CX3CL1-CX3CR1 system in these processes. Contrary to the finding of reduced microglial-dendritic shafts contacts in *Cx3cr1$^{-/-}$* mice, the proportion of spines contacted by microglia was increased in these mice. It may be suggested that this increase did not result from more putative contacts per spine, but merely reflects the fact that abGCs in *Cx3cr1$^{-/-}$* mice have less spines than in WT mice. The finding that despite the greater proportion of spines contacted in *Cx3cr1$^{-/-}$* mice there was less spine elimination indicates that microglia-spine contacts do not necessarily result in spine elimination. In fact, in developing animals it has been shown that only a small proportion of the spines that are contacted by microglia are subsequently eliminated (*Tremblay et al., 2010*). The results of the present study suggest that spine elimination following microglial contacts crucially depends on activation of the CX3CR1 by its ligand, and therefore either specific impairment of this signaling or complete microglial depletion result in reduced spine elimination. Obviously, other microglial-independent mechanisms are also involved in spine elimination (which persists to some extent also in the microglia-depleted mice).

## Possible mechanisms underlying the role of microglia in spine formation and elimination

One mechanism that may support spine formation involves the secretion of BDNF from microglia during their contacts with dendrites and spines. A recent study reported that BDNF supports the stabilization of young spines by promoting the transformation from filopodia to spines, specifically in abGCs in the OB (*Breton-Provencher et al., 2016*). The source of this BDNF may be microglial because it was previously reported that conditional microglia-specific knockout of BDNF reduced learning-associated spine formation in the motor cortex (*Parkhurst et al., 2013*). Thus, in our study the reduced numbers of microglia in the depleted mice, or the reduced contacts between microglial processes to dendrites in the *Cx3cr1$^{-/-}$* mice, may have resulted in lower secretion of BDNF near the dendrites and therefore less promotion of spine formation. The mechanism for microglia-associated synapse elimination during early development was found to involve the complement system (*Schafer et al., 2012*). Specifically, the cytokine TGFβ was found to induce expression of complement in neurons, which is sensed by complement receptors on microglia to promote phagocytosis of synapses (*Bialas and Stevens, 2013*). However, the specific role of *Cx3cr1* deficiency in these mechanisms has not been studied yet. Future research should seek to define the precise molecular mechanisms that underlie the role of microglia and their CX3CR1 signaling in the formation and elimination of abGCs synapses.

In general, the effects of *Cx3cr1* deficiency largely recapitulated the effects of microglia depletion on synaptic development of OB abGCs. Nevertheless, it should be noted that the OB micro-environment may be different in *Cx3cr1$^{-/-}$* vs. microglia-depleted mice, and thus it is possible that the similar findings involve different microglia-related mechanisms in each of the models. For example, previous studies found that CX3CR1 signaling is involved in microglial suppression and that *Cx3cr1$^{-/-}$* mice have some markers of inflammatory activation (*Wolf et al., 2013*), whereas microglia-depleted mice were found to have low levels of inflammatory activation markers (*Spangenberg et al., 2016*). In the current study, we found no differences between the two models in the OB inflammatory cytokines milieu, suggesting that in the OB the inflammatory micro-environment in these models is similar. Still, other genes are obviously differentially regulated in these models, and future research should focus on specific molecules that may mediate the proximal effects of microglial manipulations on synapse development.

It should be also noted that in each of the two models the effects of the microglia manipulation in the brain are global. Therefore, it is possible that microglia-related changes in brain regions outside the OB influence the development of abGCs in this region. Given that abGCs receive inputs only from local mitral cells in the OB, this explanation is not likely, but still the possibility that altered

inputs from other brain areas to the mitral cells indirectly influence abGCs' development cannot be ruled out. Future studies directing the microglial manipulations (depletion or *Cx3cr1* deficiency) specifically to the OB should be conducted to definitively assess this possibility.

Because both CSFR1 and CX3CR1 are expressed on peripheral macrophages and several other immune cell types (*Jung et al., 2000*; *Dai et al., 2002*), the microglia depletion and *Cx3cr1* deficiency models also involve peripheral alterations, which could indirectly influence the abGCs. Our current findings that the levels of most peripheral cytokines and chemokines were not different in the two models does not lend support for this hypothesis. Furthermore, the levels of the few peripheral molecules that did show a differential expression pattern in one of the models usually reflected a reduced inflammatory condition. For example, in the serum of $Cx3cr1^{-/-}$ mice, the levels of ICAM1 were reduced and those of CXCL12 were elevated, oppositely to the changes observed in these molecules during an inflammatory condition, such as multiple sclerosis (*Sharief et al., 1993*; *McCandless et al., 2008*). Moreover, none of the alterations in peripheral cytokines and chemokines was reflected by a parallel change in the OB. Thus, it is not likely that changes in the peripheral inflammatory environment in the two models indirectly influence brain neurons, in general, and abGCs, in particular, via peripherally-derived inflammatory molecules.

## Functional impact of microglia depletion

The dendritic spines of the OB GCs constitute the structural basis for dendro-dentritic synapses of these cells with mitral cells. These synapses are reciprocal, consisting of an excitatory (glutamatergic) mitral-to-GC synapses, adjacent to inhibitory GC-to-mitral inhibitory (GABAergic) synapses (*Egger and Urban, 2006*; *Whitman and Greer, 2007*). The latter synapses restrain MC activation in specific contexts and conditions, with important implications to olfactory functioning. In particular, these synapses contribute to lateral inhibition, which sharpens and tunes the mitral cells responses to specific odorants, intrinsic OB oscillations, which shape olfactory coding, and olfactory memory processes (*Egger and Urban, 2006*; *Abraham et al., 2010*; *Alonso et al., 2012*). Consistently with this known connectivity, the microglia depletion-induced reduction in spine density in the present study was associated with increased responsiveness of mitral cells to odors. This finding is consistent with a previous demonstration that genetically-induced impairment of the excitatory synapses between mitral cell and GCs resulted in decreased activity of the GCs, along with reduced inhibitory post-synaptic potentials in the GCs-to-mitral cells recurrent synapses (*Abraham et al., 2010*). A more dramatic manipulation of the GCs, induced by ablating OB neurogenesis altogether, resulted in reduced number of GABAergic synapses onto mitral cells, which in turn reduces the inhibitory input that these cells receive (*Breton-Provencher et al., 2009*). Obviously, in the current experiment only a relatively small subset of the MCs synapses were influenced by the microglial depletion, considering that it is estimated that about 2–10% of abGCs will be replaced during a one month period (*Ninkovic et al., 2007*). Interestingly, we found a 28.6% increase in the mitral cells responsiveness to smells, but no global effects on the response profiles of these cells (i.e., no effects on their tuning curves). Whether the decreased number of synapses of abGCs in microglia-depleted mice contributes to this increase remains an open question for future exploration.

## Conclusion

The results of the present study demonstrate for the first time that in the process of adult neurogenesis microglia do not only regulate the numbers of new-born neurons, as demonstrated previously (*Ekdahl et al., 2003*; *Monje et al., 2003*; *Butovsky et al., 2006*; *Ziv et al., 2006*; *Maggi et al., 2009*; *Sierra et al., 2010*; *Bachstetter et al., 2011*; *Reshef et al., 2014*), but that they also influence the synaptic development, maintenance, dynamics and functioning of adult-born neurons. These findings are consistent with previous reports regarding the role of microglia in synapse pruning, plasticity and circuits formation during brain development (*Paolicelli et al., 2011*; *Paolicelli and Gross, 2011*; *Schafer et al., 2012*; *Zhan et al., 2014*), in neuroplasticity induced by environmental enrichment (*Maggi et al., 2011*; *Reshef et al., 2014*) and in learning-dependent spine formation in the adult brain (*Parkhurst et al., 2013*). Together, these findings indicate that microglia have a general role in modulation of neuronal plasticity and its involvement in neuronal circuit connectivity throughout the entire life span of animals. Accordingly, during various pathological conditions, including neuroinflammation, neurodegeneration, neurodevelopmental and neuropsychiatric diseases, at least

some of the symptoms may result from direct detrimental effects of the microglial disturbances that are associated with these conditions on neural plasticity and circuit remodeling processes (*Prinz and Priller, 2014*; *Salter and Beggs, 2014*; *Zhan et al., 2014*; *Chung et al., 2015*; *Yirmiya et al., 2015*).

# Materials and methods

**Key resources table**

| Reagent type (species) or resource | Designation | Source or reference | Identifiers | Additional information |
|---|---|---|---|---|
| strain, strain background (Mice) | *Cx3cr1$^{-/-}$* | The Jackson Laboratories | B6.129P-Cx3cr1tm1Litt/J. (RRID:IMSR_JAX:005582) | On C57BL/6 background (serving as wild-type control) |
| strain, strain background (Mice) | Thy1-GCaMP3 | Kindly provided to Adi Mizrahi from Guoping Feng. *Chen et al. (2012)* | Available in Jackson as B6;CBA-Tg(Thy1-GCaMP3)6Gfng/J. (RRID:IMSR_JAX:017893) | On C57BL/6 background (serving as wild-type control) |
| antibody | Anti-Iba-1, Rabbit | Wako, Osaka, Japan | Wako Cat. No. 019–19741. (RRID:AB_2665520) | 1:1000, in blocker solution. Shaker, Room temp |
| antibody | Anti-DCX, Guinea pig | Millipore (Mercury), CA, U.S.A. | Millipore Cat. No. AB2253. (RRID:AB_2230227) | 1:1000, in blocker solution. Shaker, Room temp |
| antibody | Anti-BrdU, Rat | Harlan Sera-Lab, Loughborough, U.K. | Harlan-Sera Cat. No. OBT0030 (RRID:AB_2314037) | 1:200, in blocker solution. Shaker, Room temp |
| antibody | Anti-NeuN, Rabbit | Cell signaling, MA, U.S.A. | Cell Signaling Cat. No. 24307 (RRID:AB_2651140) | 1:200, in blocker solution. Shaker, Room temp |
| antibody | Anti-RFP, Rabbit | Rockland, PA, U.S.A. | Rockland Cat. No. 600-401-379 (RRID:AB_2209751) | 1:1000, in blocker solution. Shaker, Room temp |
| recombinant DNA reagent | Details for AAV1 | Upenn, PA, U.S.A. | Upenn viral core Cat. No. AV-1-PV3365 | AAV1 under the CAG promoter, expressing TdTomato |
| commercial assay or kit | Cytokine and chemokine microarray assay | R and D Systems, MN, U.S.A. | Proteome Profiler Mouse Cytokine Array Kit (ARY006) | |
| c chemical compound, drug | PLX5622 | PLEXXIKON Inc., CA, U.S.A. | AIN-76A Rodent Diet with PLX5622 | 1,200 mg PLX5622 (Free Base)/kg |
| chemical compound, drug | Control diet | PLEXXIKON Inc., CA, U.S.A. | AIN-76A Rodent Diet | |
| chemical compound, drug | BrdU | Sigma-Aldrich, MO, U.S.A. | Sigma-Aldrich Cat. NO. B5002 | 10 mg/ml IP injection |
| software, algorithm | ImageJ/Figi | University of Wisconsin, Madison, WI, U.S.A. | (http://rsb.info.nih.gov/ij/), (http://fiji.sc/Fiji) (RRID:SCR_003070) | |
| software, algorithm | Matlab | Mathworks Inc., U.S.A. | Matlab, R2014a (RRID:SCR_001622) | |
| software, algorithm | SPSS | I.B.M., U.S.A. | SPSS, Version 19 (RRID:SCR_002865) | |

## Animals

Animals were 10–12 week old male C57BL/6 mice, *Cx3cr1$^{-/-}$* mice, in which both copies of the *Cx3cr1* gene were replaced by a green fluorescent protein (GFP) reporter gene, resulting in deletion of the *Cx3cr1* gene (*Jung et al., 2000*), or Thy1-GCaMP3 mice, expressing the GFP-based calcium sensor GCaMP, which provides a powerful tool for detecting neuronal activity-related calcium transients, under the control of the Thy1 promoter (*Chen et al., 2012*). Both mutant strains were generated on the C57BL/6 genetic background, so mice from this strain served as wild type (WT) controls in the relevant experiments. All experiments were approved by the Hebrew University Ethics Committee on Animal Care and Use.

## Microglia depletion

Under normal conditions, microglia are the only brain cell type that expresses colony stimulating factor one receptors (CSF1R), whose signaling is essential for microglial proliferation, differentiation

and survival. Indeed, systemic treatment with the selective CSF1 receptor kinase inhibitor PLX3397 was recently shown to deplete ~99% of brain microglia cells, already after 7 days of treatment, with no effects on the number and morphology of neurons or astrocytes (*Elmore et al., 2014*). In the present study, we administered a similar CSF1R antagonist (PLX5622 1200 mg/kg) (*Dagher et al., 2015*) via the diet (i.e. food pellets) for 4 weeks, beginning three days after the virus injections as described below. Mice fed an identical diet with no PLX5622 served as controls. The PLX5622 diet was generously contributed by Plexxikon inc. (Berkeley, USA). Mice ate their regular daily amount of chow, which was either supplemented or not with PLX5622.

## Virus injections

To label adult-born neurons, we injected an adeno-associated virus with a number one serotype (AAV1) encoding TdTomato into the rostral migratory stream (RMS), as described previously (*Nissant et al., 2009*; *Livneh and Mizrahi, 2012*). The AAV1 was ordered from UPENN viral core, and it expressed TdTomato under the control of the CAG promoter. Virus injections into the RMS were performed as previously described (*Livneh and Mizrahi, 2012*). Mice were anesthetized using ketamine (100 mg per kg of body weight) and medetomidine (0.83 mg per kg of body weight), with carprofen (4 mg per kg of body weight). Saline was injected subcutaneously to prevent dehydration. Depth of anesthesia was assessed by monitoring the pinch withdrawal reflex. Injections were done stereotaxically using hydraulic pressure (coordinates relative to Bregma: anterior- 3.3 mm, lateral- 0.8 mm, ventral- 2.9 mm). A chronic window was implanted immediately after the virus injection as described in the next passage.

## Chronic window implantation over the olfactory bulb (OB)

For the time-lapse imaging experiment of adult-born granule cells (abGCs) and calcium imaging mitral cells (MCs) we used a cranial window preparation, which was similar to the one recently described in detail (*Adam and Mizrahi, 2011*). Briefly, immediately after the RMS virus injection, the skull overlying one of the OBs was carefully removed using a micro-driller, leaving the dura intact. Then, a custom-made rectangular glass coverslip (No. 2) was positioned over the opening, and 1.5% agarose was used to cover exposed brain tissue, if present. The window was then sealed in place using histoacryl (TissueSeal) and dental cement. A 0.1 g metal bar was glued to the skull for repositioning the animal's head under the microscope in consecutive imaging sessions (*Mizrahi and Katz, 2003*).

After surgery, mice were allowed to fully recover for several hours and returned to the animal facility under normal housing conditions. Approximately 28 days later, mice were anesthetized and underwent the first imaging session. It should be noted that at this time microglia in $Cx3cr1^{-/-}$ mice were not activated, as evidenced by immunohistological examination comparing the number and morphology of microglia in the OB of the implanted hemisphere as compared to the OB from the non-implanted hemisphere (data not sown).

## In vivo two-photon imaging

Time-lapse imaging of new-born GCs started 27 days post-injection (d.p.i), and was performed again at 28 d.p.i. (i.e., a 24 hr interval). Mice were anesthetized (using ketamine (50 mg per kg)/medetomidine (0.42 mg per kg)) and placed under the microscope in a custom-made stereotaxic device via the metal bar glued to the skull in a fixed orientation relative to the objective lens. We performed the imaging of the OB using an Ultima two-photon microscope from Prairie Technologies (Middleton, WI), equipped with a 16X water-immersion objective lens (0.8 NA; CF175, Nikon). We delivered two-photon excitation (1000 nm) with a DeepSee femtosecond laser (Spectraphysics), and expanded the laser beam to fill the large back aperture of the 16X objective. We acquired images of dendritic spines (512 × 512 pixels) at 0.23 μm/pixel resolution in the xy dimension and at 0.9 μm/frame in the z dimension. Each dendritic tree was identified in the additional imaging sessions by its location in 3D relative to the blood vessel map (*Mizrahi, 2007*; *Kopel et al., 2012*).

## In vivo two-photon calcium imaging

Mice were imaged under ketamine/medetomidine anesthesia (100 mg/kg and 0.83 mg/kg, respectively, i.p.). Imaging was conducted with an Ultima two-photon microscope from Prairie

Technologies (Middleton, WI, USA). GCaMP3 was excited at 925 nm (using the DeepSee femtosecond laser from Spectra-physics) and images (420 × 190 pixels) were acquired at 7 Hz, using a 16x objective (NIKON). Imaging cycles were triggered by the odor delivery system. The odor presentation protocol included 8 stimuli of monomolecular odors+ blank. Odor presentation was repeated three times in each imaging field. For each trial we collected 10 s before the stimulus onset and the total trial duration was 22 s. On the first imaging session, several fields of view (3-5) were imaged in each mouse for a total of 150–200 MCs per mouse. After the first imaging session, animals were initiated on a PLX5622 diet for the next 4 weeks. 28 days after the first imaging session we reimaged all mice, using the same odor stimulation protocol. In a few fields we could reliably identify the same neurons from the first imaging session.

## Odor delivery

Eight monomolecular odorants were obtained from Sigma Aldrich (ethyl-tiglate, ethyl-butyrate, Isoamyl acetate, amyl acetate, geraniol, eucalyptol, α-pinnen, 2 Phenylethanol). For the odor stimulus presentation, we used a nine-odor air dilution olfactometer (RP Metrix Scalable Olfactometer Module LASOM 2), as described by others (*Smear et al., 2013*). Briefly, the odorants were diluted in the mineral oil to 10 ppm concentration. Saturated vapor was obtained by flowing nitrogen gas at flow rates of 100 ml/min through the vial with the liquid odorant. The odor streams were mixed with clean air adjusted to produce a constant final flow rate of 900 ml/min. Odors were further diluted tenfold before reaching the final valve (via a four-way Teflon valve, NResearch). In between stimuli, 1000 ml/min of a steady stream of filtered clean air flowed to the odor port continuously. During stimulus delivery, a final valve switched the odor flow to the odor port, and diverted the clean airflow to the exhaust. Odors were delivered at a flow rate of 1 L/min for 2 s. Inter-trial interval was 30 s. All flows were adjusted to minimize the pressure shock resulting from line switching stabilization after opening the final valve. The olfactometer was calibrated using a miniPID (Aurora Scientific).

## BrdU administration

5-bromo-2'- deoxyuridine (BrdU; 10 mg/ml; Sigma, St. Louis, MO), a marker of cell proliferation, was administered i.p. (0.1 mg/g of body weight). Mice received four daily injections of BrdU on both day 28 and day 27 before the termination of the experiment (i.e., 2–3 days before the initiation of the diet treatment). Because during the injection period the proliferation of new-born cells was comparable between the groups, the quantification of BrdU-labeled cells after the experiment provides a measure of their survival rate.

## Immunohistochemistry

At d.p.i 28, mice were perfused transcardially with PBS, followed by 4% paraformaldehyde, and the brains were cryoprotected in 30% sucrose overnight. OBs were sectioned coronally (40 μm) on a sliding microtome. Free floating slices were washed in PBS and then incubated for 1 hr in a blocking solution (5% normal goat serum and 0.5% Triton-x in PBS). Slices were incubated overnight at room temperature with primary antibodies diluted in the blocking solution (rabbit anti-RFP, 1:1000, Rockland; rabbit-anti Iba-1, 1:1000, Wako; guinea pig anti-DCX, 1:1000, Millipore), and then with a secondary antibody, also in a blocking solution (goat anti-rabbit-conjugated-Cy3, 1:500, Jackson ImmunoResearch; goat anti-rabbit-conjugated-Cy5, 1:500, Jackson ImmunoResearch; biotin-SP-conjugated donkey anti-guinea pig, 1:200; Jackson) for 3 hr. For DCX staining only, slices were then incubated with a tertiary antibody for 1 hr at RT (conjugated streptavidin, 1:200, Jackson).

For quantification of BrdU-labeled neurons, sections were double-labeled for BrdU and the neuronal marker NeuN. Slides were first pretreated by DNA denaturation (50% Formamide/50% 2XSSC for 120 min at 65°C; and then in 2 N HCl for 30 min at 37°C), and later incubated for 48 hr at 4°C with the primary antibodies: rat monoclonal anti-BrdU antibody (1:200; Accurate Scientific, Harlan Sera-Lab, UK) and rabbit monoclonal anti-NeuN antibody (1:200; Cell signaling, USA). Sections were then incubated with a secondary antibody (donkey anti rat IgG conjugated to Cy5 (647), Jackson laboratories; and goat anti-rabbit IgG conjugated to Cy3 (555), obtained from Alexa (both at 1:250)).

For double staining of both abGCs and microglia, we stained the abGCs with the TdTomato antibody (rabbit anti-RFP, 1:1000, Rockland) and the secondary antibody (goat anti-rabbit-conjugated-Cy3, 1:500, Jackson ImmunoResearch). The slices were blocked with 5% normal rabbit serum and

0.5% Triton-x in PBS and then with Affinipure Fab fragment Goat anti-Rabbit, 1:750, ImmunoResearch. Microglia were then stained with the first antibody (rabbit-anti Iba-1, 1:1000, Wako) and the secondary antibody (goat anti-rabbit-conjugated-Cy5, 1:500, Jackson ImmunoResearch). Prior to mounting on microscope slides, we incubated slides with DAPI (Santa Cruz Biotechnology; 50 μg/ml) for 5 min and then washed them with PBS.

## Confocal microscopy

Slices were imaged at 0.165–0.2 μm/pixel in the XY dimension and at 0.5 μm steps in the Z dimension. Imaging was performed with an Olympus FV-1000 confocal microscope, via a 10X (0.4 NA), a 20X (0.8 NA) and a 40X (1.3 NA) oil objective or with a Leica SP5 confocal microscope via a 40X (1.25 NA) oil objective. We imaged and analyzed only distal abGCs dendrites in the external plexiform layer (EPL), as different parts of the GC dendritic tree were shown to have different properties (*Kelsch et al., 2008*). tdTomato expression levels were variable between neurons. Iba-1 staining in microglia was also variable. Thus, we limited our analysis to the brightest neurons and microglia in the slice. This bias to high contrast signal may be one reason for the higher average spine densities reported here, compared with what we and others have reported earlier (*Whitman and Greer, 2007*; *Dahlen et al., 2011*; *Livneh and Mizrahi, 2011*) (but see also [*Breton-Provencher et al., 2009*; *Lin et al., 2010*]). Another possible reason for the higher average spine densities is the way we scored spines with multiple heads (see below, Data analysis).

## Morphological data analysis

All analyses in this study were done by an experimenter blind to the experimental condition. Quantitative analyses of spine dynamics were performed manually from the filtered image stacks (Gaussian blur filter), using ImageJ (http://rsb.info.nih.gov/ij/). The imaged regions in the two sessions were aligned to each other using the image registration and 'Sync windows' plugin (http://rsb.info.nih.gov/ij/plugins/sync-windows.html). Each spine was then scored as stable, lost, or gained, using the 'cell counter' plugin (http://rsb.info.nih.gov/ij/plugins/cell-counter.html). For each cell, spine dynamics were calculated as follows: number of stable spines = Nstable/(Nstable + Nlost), number of lost spines = Nlost/(Nstable + Nlost), number of gained spines = Ngained/(Nstable + Ngained). Nstable, Nlost, and Ngained are the number of stable, lost, and gained spines, respectively. Dendrites and spines were analyzed only if they had strong fluorescent signal that was clearly contrasted from the background. To estimate the size of a spine, we used the integrated fluorescence of a spine head, which has been shown to be a good proxy of its size (*Holtmaat et al., 2005*). Analysis was performed by using custom software written in Matlab (MathWorks)(*Source code 1*). First, each frame of the stack was filtered with a $3 \times 3$ pixel median filter. Then, for each spine, the head, adjacent background, and shaft regions were marked manually. Finally, spine head size was calculated in two dimensions at the image frame in which the integrated fluorescence of a single spine head was highest. Intensity values of all pixels comprising the spine head were summed after subtracting the mean value of the background adjacent to the spine. To normalize this value and to avoid biases that can be caused by inhomogeneities in excitation level and TdTomato expression, the integrated value of the fluorescence of a single spine head was divided by averaging the intensity levels of the pixels on the adjacent dendritic shaft, whose intensity level was on the upper 10% of that dendrite. The result of this analysis was the spine head size having arbitrary units (AU). Higher values of this measure reflected bigger spine head sizes. In the current dataset, spines ranged from 0.15 to 70 AU.

tdTomato-based spine density was calculated from arbitrarily chosen dendritic trees within the EPL, belonging to the same neuron. Dendritic length and spine numbers were calculated using Fiji image processing package (http://fiji.sc/Fiji), with the use of the 'simple neurite tracer' and the 'cell counter' plugins. In several cases we observed multiple spines emerging from the same spine neck or parent dendrite. In those cases, all spine heads were counted as spines, regardless of the neck origin.

Microglial putative contacts with abGCs were calculated from arbitrarily chosen dendritic trees within the EPL, belonging to the same neuron. Dendritic length and putative contacts were calculated from image Z stacks using Fiji image processing package, with the use of the 'simple neurite tracer' and the 'cell counter' plugins. A putative dendritic contact was defined as double staining (marked by a yellow color) of a microglia process (Cy5) and an abGC dendritic shaft (TdTomato)

(*Videos 1*, *2*). The number of contacts per dendritc shaft was normalized, by dividing this number by the dendritic shaft's length. A putative spine contact was defined as double staining (yellow staining) of a microglial process and an abGC dendritic spine. For each abGC we calculated the number of spines that were contacted by microglial processes out of the total spine number.

The microglial morphological analysis was conducted on image stacks, using Fiji (http://fiji.sc/Fiji), with the use of the 'simple neurite tracer' plugin.

The total numbers of Iba-1-labeled microglia and DCX- labeled newborn neurons were estimated using images taken by X10 lens, from the first 8 µm out of each 40 µm slice. In each slice, the number of Iba-1- or DCX-labeled cells (that were also counter-stained with DAPI to ascertain nuclear staining) were manually counted in a defined area exclusively containing the entire olfactory bulb (including the glomerular, plexiform, mitral and granule layers), using the ImageJ cell counter tool. For each brain, microglia and newborn neurons number as well as microglial morphology were assessed in eight sections along the entire rostro-caudal axis of the olfactory bulb.

The total numbers of Brdu-NeuN-labeled newborn neurons were estimated using images taken by X20 lens, from the first 8 µm out of each 40 µm slice. In each slice, the number of Brdu-NeuN-labeled cells were manually counted in a defined area of the exclusively containing the granule cell layer, using the imagej cell counter tool. Only cells co-labeled with NeuN and Brdu along the Z-axis were counted. For each brain, newborn neurons numbers were assessed in eight sections along the entire rostro-caudal axis of the olfactory bulb.

## Calcium imaging data analysis

Data was analyzed using ImageJ, followed by a custom code written in MATLAB (The Mathworks). ImageJ was used in order to manually mark regions of interest (ROIs) corresponding to individual cell bodies and to extract the mean fluorescence for each cell body at any given point in time. Thus, the extracted data was constructed from a vector of 164 values (representing the mean fluorescence in the ROI over time) for each single trial. Each trial was composed of 5 s pre-stimulation, 2 s odor presentation, and 5 s post stimulation for a total of 14 s (i.e.,164 time points at 7 Hz). Each trial was separately smoothed with the MATLAB 'smooth' function using a moving average filter with a span of 5. Relative fluorescence change ($\Delta F/F$) was calculated using the mean fluorescence over 4 s before odor onset as the baseline fluorescence (F0). We defined a response window equal to the stimulus duration +3 s. In order to determine whether a response was statistically significant, we created a distribution of Z scores based on the statistics of each trial separately as follows: ZResponse = Response magnitude/Trial noise, where Response magnitude = peak $\Delta F/F$ during response window - peak $\Delta F/F$ during the parallel time window in the 'Blank' channel. Trial noise is the standard deviation during the response window at 'Blank'. We used a threshold of ZResponse = 2 to define a response as significant. Notably, ZResponse thresholds ranging from 1.5 to 3, all yielded qualitatively similar results. Lastly, we categorized calcium transients as odor evoked responses if at least two trials in addition to the mean trace had Z scores that surpassed the threshold. The dataset underwent two different analyses – whole population and time-lapse.

For the whole population analysis, we used all recorded cells imaged at either time points and performed unpaired statistics. Neurons showing responses at the blank channel above 0.3 $\Delta F/F$ as well as neurons with seemingly artifactual changes in >0.5 $\Delta F/F$ before odor presentation were discarded. Inclusion of these cells in the analysis did not qualitatively change the results. To assess changes in response profile we quantified two values: (1) the mean number of odors eliciting a response per cell, and (2) the magnitude of all responses.

For the time lapse analysis, we used only cells that were clearly identified at both imaging sessions. Thus, these neurons were also analyzed in the whole population analysis. The statistics for this part of the analysis was based on paired samples comparisons. For the scatter plot we used only cell-odor pairs that showed a significant response at least at one of the imaging sessions.

## RNA sequencing

Mice were sacrificed by decapitation. Brains were quickly removed on an ice-cold glass plate, and the olfactory bulbs were dissected out. Tissues were weighed, flash frozen in liquid nitrogen, and stored at −80℃. RNA was extracted using PerfectPure RNA extraction kit (5 PRIME, Darmstadt,

Germany) and RNA samples (2 µg) were reverse transcribed using the QuantiTect Reverse Transcription Kit from Qiagen (Hilden, Germany), including DNase treatment of contaminating genomic DNA.

For RNA Sequencing, PolyA based mRNA was selected using oligodT beads, followed by fragmentation, first strand and second strand synthesis reactions. Illumina libraries were constructed while performing the end repair, A base addition, adapter ligation and PCR amplification steps with SPRI beads cleanup in between steps. Indexed samples were pooled and sequenced in an Illumina HiSeq 2500 machine in a single read mode.

### Bioinformatics analysis

Adapters were trimmed using the cutadapt tool. Following adapter removal, reads that were shorter than 40 nucleotides were discarded (cutadapt option –m 40). Reads that had either a percentage of Adenine bases above 50% or a percentage of Thymine bases above 50% were discarded using a custom script. TopHat (v2.0.10) was used to align the reads to the mouse genome (mm10) (*Kim et al., 2013*). Counting reads on mm10 refseq genes (downloaded from igenomes) was done with HTSeq-count (version 0.6.1p1) (*Anders et al., 2015*). Differential expression analysis was performed using DESeq2 (1.6.3) (*Anders et al., 2013*; *Love et al., 2014*). Raw p values were adjusted for multiple testing using the procedure of Benjamini and Hochberg (*Benjamini et al., 2001*)

### Cytokine microarray

OB's were excised and homogenized in PBS with protease inhibitor cocktail (Sigma) and 1% Triton X-100. The amount of extracted proteins was measured both by NanoDrop and by a Pierce BCA protein assay kit (Thermo Scientific). Trunk blood was collected following brief isoflurane anesthesia and decapitation. The blood was allowed to clot for 30 mins at room temperature and then centrifuged for 10 mins at 1500 g for serum collection. Cytokine and chemokine measurements were carried out using Proteome Profiler Mouse Chemokine and Cytokine Array (R&D Systems) according to the manufacturer's instructions. Cytokine and chemokine expression levels were measured by densitometry, and expressed as arbitrary units relative to the assay's internal control.

### Statistical analysis

All data are presented as mean ±S.E.M. Sample sizes were determined in advance. Based on previous research in our and other laboratories, examining synaptic density and dynamics in adult-born neurons, we analyzed about 18 neurons (range: 13–20) from four mice in each experimental and control group. This sample size was previously found to have sufficient statistical power to demonstrate the effects of various developmental, genetic and environmental factors. Animals were allocated into the experimental/control groups in a randomized manner. Following the randomization, we ascertained that the different experimental/control groups were matched for age and body weight, and adjusted the allocation if needed. Data was tested for normality using the Lilliefors test, and then compared using two-tailed Student's t-test (unless noted). For relatively large samples (n > 30), t-tests were applied also to distributions that did not pass the normality test. Spine turnover data was analyzed by mixed ANOVA, followed by specific comparisons with Bonferonni's corrections. Histograms of spine sizes, proportions of responsive Mitral cells and response magnitudes were tested using the two-sampled Kolmogorov-Smirnov test. RNA-seq analysis was done using binomial linear models (DSEq2 algorithm).

### Acknowledgements

We thank Mr. Nitai Harari and Ms. Hagar Lev-Ari for help with running the experiments, and Ms. Zehava Cohen for help in preparation of the figures. We thank Gilgi Friedlander, as well as the The Mantoux Bioinformatics institute of the Nancy and Stephen Grand Israel National Center for Personalized Medicine (INCMP) for their assistance in DNA sequencing. We thank Plexxikon Inc. (Berkeley, USA) for the generous contribution of PLX5622 diet. This research was supported by the ISRAEL SCIENCE FOUNDATION – FIRST Program grant number 1357/13, by the ISRAEL SCIENCE FOUNDATION grant No. 1379/16 (to RY), and by the ISRAEL SCIENCE FOUNDATION grant number 1284/10 (to AM).

## Additional information

### Funding

| Funder | Grant reference number | Author |
| --- | --- | --- |
| Israel Science Foundation | FIRST Program. 1357/13 | Raz Yirmiya |
| Israel Science Foundation | Grant No. 1379/16 | Raz Yirmiya |
| Israel Science Foundation | Grant No. 1284/10 | Adi Mizrahi |

The funders had no role in study design, data collection and interpretation, or the decision to submit the work for publication.

### Author contributions
Ronen Reshef, Conceptualization, Data curation, Software, Formal analysis, Validation, Investigation, Visualization, Methodology, Writing—original draft, Project administration, Writing—review and editing; Elena Kudryavitskaya, Investigation, Carried out experiments; Haran Shani-Narkiss, Formal analysis, Visualization; Batya Isaacson, Investigation, Performed experiments; Neta Rimmerman, Formal analysis, Investigation, Visualization; Adi Mizrahi, Conceptualization, Supervision, Funding acquisition, Visualization, Methodology, Writing—review and editing; Raz Yirmiya, Conceptualization, Supervision, Funding acquisition, Visualization, Methodology, Writing—original draft, Project administration, Writing—review and editing

### Author ORCIDs
Ronen Reshef (iD) http://orcid.org/0000-0003-3970-6687
Raz Yirmiya (iD) http://orcid.org/0000-0002-4009-7316

### Ethics
Animal experimentation: All experiments were approved by the Hebrew University Ethics Committee on Animal Care and Use (IACUC protocols11-12857-4)

### Decision letter and Author response
Decision letter https://doi.org/10.7554/eLife.30809.031
Author response https://doi.org/10.7554/eLife.30809.032

## Additional files

### Supplementary files
• Supplementary file 1. Gene transcripts significantly differentially regulated in the olfactory bulb of PLX5622-treated mice, compared with control diet-treated mice.
DOI: https://doi.org/10.7554/eLife.30809.024

• Supplementary file 2. Comparisons between gene transcript expression of 40 cytokines and chemokines in the olfactory bulbs of $Cx3cr1^{-/-}$ (KO) and PLX5622-treated (PLX) mice, as well as their respective wild type (WT) and control diet-treated (CON) mice.
DOI: https://doi.org/10.7554/eLife.30809.025

• Source code 1. Calculation of spine size from flourscent Z stacks.
DOI: https://doi.org/10.7554/eLife.30809.026

• Transparent reporting form
DOI: https://doi.org/10.7554/eLife.30809.027

### Major datasets
The following dataset was generated:

| Author(s) | Year | Dataset title | Dataset URL | Database, license, and accessibility information |
|---|---|---|---|---|
| Reshef R, Kudrya-vitskaya E, Shani-Narkiss H, Isaacson B, Rimmerman N, Mizrahi A, Yirmiya R | 2017 | The role of microglia in maturation of adult-born neurons | https://www.ncbi.nlm.nih.gov/geo/query/acc.cgi?acc=GSE106857 | Publicly available at the NCBI Gene Expression Omnibus (accession no: GSE106857) |

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
