## [Decision Letter]

Thank you for submitting your article "The role of microglia and their CX3CR1 signaling in adult neurogenesis in the olfactory bulb" for consideration by *eLife*. Your article has been reviewed by three peer reviewers, and the evaluation has been overseen by a Reviewing Editor and Marianne Bronner as the Senior Editor. The reviewers have opted to remain anonymous.

The reviewers have discussed the reviews with one another and the Reviewing Editor has drafted this decision to help you prepare a revised submission.

Summary:

This paper represents the first demonstration that microglia are the key player in spine pruning. The results are interesting because both elimination and formation of synapses is impacted by microglia depletion. Moreover, depletion of CX3CR1 from microglia, which results in abnormal neuron-microglia communication, led to similar results as microglia depletion, namely impaired synaptic running/development.

Essential revisions:

1) Along the same lines, depletion of microglia versus the use of microglia deficient in CX3CR1 create distinct micro environments in the brain, and therefore these two animal models should be carefully compared. CX3CR1 is a receptor through which the neurons suppress microglial activity. Therefore in Cx3CR1-/- mice the microenvironment is more pro-inflammatory, and could have a major impact on the outcome. The authors should monitor levels of pro-inflammatory cytokines in the OB in these two animal models and address this issue.

2) In Figure 3, the effect of microglial depletion in adulthood should be verified and documented. Is it as effective as the same manipulation at earlier stages? Staining for microglia is needed.

3) The RNAseq results as presented, and conclusion do not add to the story; these results only confirm that microglia were depleted. It would be more helpful to thoroughly compare the cytokine milieu.

4) Overall the discussion is somewhat confusing, in light of the fact that the two model systems that create completely distinct conditions. This is not sufficiently clarified in the text.

---

## [Author Response]

Essential revisions:1) Along the same lines, depletion of microglia versus the use of microglia deficient in CX3CR1 create distinct micro environments in the brain, and therefore these two animal models should be carefully compared. CX3CR1 is a receptor through which the neurons suppress microglial activity. Therefore in Cx3CR1-/- mice the microenvironment is more pro-inflammatory, and could have a major impact on the outcome. The authors should monitor levels of pro-inflammatory cytokines in the OB in these two animal models and address this issue.

We agree with the reviewer that the two models create distinct micro environments in the brain. Differences in the inflammatory environment could certainly contribute to the changes that were observed, and therefore, as suggested by the reviewer, we compared the cytokine and chemokine milieus in the two models. We used two methods for this comparison. Firstly, we conducted an RNA-seq analysis, measuring the expression levels of 40 inflammatory cytokines and chemokines in the OB of PLX5622-treated, *Cx3cr1^-/-^* and their respective control mice. Secondly, we conducted a microarray assay, comparing the protein levels of the same 40 inflammatory molecules in these groups. Both analyses indicated that there were no differences between the groups in the levels of olfactory bulb pro-inflammatory cytokines and chemokines (e.g., IL-1β, TNFα, IL-6 and IFNγ), either at the RNA transcription or the protein expression levels. These findings are now mentioned in the Results section (subsection “Comparison of the cytokine/chemokine expression profile between the two microglia manipulation models”, Supplementary file 2, and Figure 9). In the Discussion section, the distinctiveness of the micro environments created in the two models is acknowledged and discussed (subsection “Possible mechanisms underlying the role of microglia in spine formation and elimination”, as well as the response to point 4 below).

2) In Figure 3, the effect of microglial depletion in adulthood should be verified and documented. Is it as effective as the same manipulation at earlier stages? Staining for microglia is needed.

As suggested by the Reviewer, we conducted an additional experiment assessing the effects of microglia depletion on mature GCs. In this experiment, we injected tdTomato-encoding AAV1 into the RMS, treating the animals with the PLX5622 diet for 25 days, from 45 to 70 days post-injection, i.e., when the GCs were already mature (Figure 3). This treatment reduced the numbers of microglia dramatically (Figure 3—figure supplement 1), similarly to the effect shown above for animals treated with PLX5622 at 3 to 28 days post-injection. However, following the delayed microglia depletion induction there was no difference between the spine density in the PLX5622-treated and the control groups (Figure 3). These findings are described in the Results section (third paragraph subsection “Microglia depletion reduces spine density in adult-born neurons”).

3) The RNAseq results as presented, and conclusion do not add to the story; these results only confirm that microglia were depleted. It would be more helpful to thoroughly compare the cytokine milieu.

We agree with the reviewer that the RNAseq results as presented do not add to the story. Therefore, we mention the results of the RNA-Seq analysis in PLX5622-treated mice only as a confirmation of the microglia depletion (first paragraph subsection “Microglia depletion reduces spine density in adult-born neurons”), whereas the IPA pathway analysis was removed from the manuscript. As suggested, we added a thorough comparison of the cytokine milieu in the two models (see comment 1 above, in Essential Revision).

4) Overall the discussion is somewhat confusing, in light of the fact that the two model systems that create completely distinct conditions. This is not sufficiently clarified in the text.

In accordance with the Reviewer’s comment we thoroughly revised the Discussion section, separating the discussion of the two microglia manipulation models. We also addressed the distinctiveness of the two models in the discussion, asserting that: “In general, the effects of *Cx3cr1^-/-^* deficiency largely recapitulated the effects of microglia depletion on synaptic development of OB abGCs. Nevertheless, it should be noted that the OB micro-environment may be different in *Cx3cr1^-/-^* vs. microglia-depleted mice, and thus it is possible that the similar findings involve different microglia-related mechanisms in each of the models. For example, previous studies found that CX3CR1 signaling is involved in microglial suppression and that *Cx3cr1^-/-^* mice have some markers of inflammatory activation (Wolf et al., 2013), whereas microglia-depleted mice were found to have low levels of inflammatory activation markers (Spangenberg et al., 2016). In the current study, we found no differences between the two models in the OB inflammatory cytokines milieu, suggesting that in the OB the inflammatory micro-environment in these models is similar. Still, other genes are obviously differentially regulated in these models, and future research should focus on specific molecules that may mediate the proximal effects of microglial manipulations on synapse development” (subsection “Possible mechanisms underlying the role of microglia in spine formation and elimination”).